# Learning to reconstruct from saturated data: audio declipping and high-dynamic range imaging

**Victor Sechaud**                                    *victor.sechaud@ens-lyon.fr*
*CNRS and ENS de Lyon, Laboratoire de physique,*
*Lyon, France*

**Laurent Jacques**                                  *laurent.jacques@uclouvain.be*
*UCLouvain, ICTEAM,*
*Louvain-la-Neuve, Belgium*

**Patrice Abry**                                       *patrice.abry@ens-lyon.fr*
*CNRS and ENS de Lyon, Laboratoire de physique,*
*Lyon, France CIFAR Fellow*

**Julián Tachella**                                    *julian.tachella@ens-lyon.fr*
*CNRS and ENS de Lyon, Laboratoire de physique,*
*Lyon, France*

**Reviewed on OpenReview:** *https://openreview.net/forum?id=AkJWgglkLd*

## Abstract

Learning based methods are now ubiquitous for solving inverse problems, but their deployment in real-world applications is often hindered by the lack of ground truth references for training. Recent self-supervised learning strategies offer a promising alternative, avoiding the need for ground truth. However, most existing methods are limited to linear inverse problems. This work extends self-supervised learning to the non-linear problem of recovering audio and images from clipped measurements, by assuming that the signal distribution is approximately invariant to changes in amplitude. We provide sufficient conditions for learning to reconstruct from saturated signals alone and a self-supervised loss that can be used to train reconstruction networks. Experiments on both audio and image data show that the proposed approach is almost as effective as fully supervised approaches, despite relying solely on clipped measurements for training.

## 1 Introduction

Inverse problems appear in various engineering and science applications, such as tomography (Jin et al., 2016), MRI (Lustig et al., 2007) or phase retrieval (Shechtman et al., 2015). They are described by the following forward equation:

$$\boldsymbol{y} = h(\boldsymbol{x}) + \boldsymbol{\epsilon}, \tag{1}$$

where the aim is to recover the ground-truth $\boldsymbol{x} \in \mathcal{X} \subseteq \mathbb{R}^n$ from measurements $\boldsymbol{y} \in \mathcal{Y} \subseteq \mathbb{R}^m$. The set $\mathcal{X}$ defines the support of the of the signal distribution $p(\boldsymbol{x})$, which we refer to as the "signal set". The function $h : \mathbb{R}^n \mapsto \mathbb{R}^m$ represents the forward operator and $\boldsymbol{\epsilon} \in \mathbb{R}^m$ the noise. These problems are often ill-posed due to potential information loss induced by $h$ (for example, if $m < n$) or the existence of an infinite number of solutions. One way of solving this is to impose prior information on $\boldsymbol{x}$, such as piecewise constant assumptions (Rudin et al., 1992). However, choosing the right prior can be challenging and misspecified priors can introduce biases or give poor approximations of the underlying signals. This problem can be mitigated by learning the prior or reconstruction function directly from data. Most learning based methods learn an inverse operator of $h$ on $\mathcal{X}$, $f : \boldsymbol{y} \mapsto \boldsymbol{x}$ from a training dataset containing ground truth with their

associated measurements $\{(\boldsymbol{x}_i, \boldsymbol{y}_i)\}_{i=1}^{N}$ and a neural network $f_{\boldsymbol{\theta}}$ that parameterizes the inverse function. The standard approach computes the parameters $\boldsymbol{\theta} \in \mathbb{R}^p$ by minimizing the mean square error (MSE):

$$\hat{\boldsymbol{\theta}} = \arg \min_{\boldsymbol{\theta}} \sum_{i=1}^{N} \|f_{\boldsymbol{\theta}}(\boldsymbol{y}_i) - \boldsymbol{x}_i\|^2. \tag{2}$$

Despite the good performance in some applications, this approach suffers from two main drawbacks: *(i)* ground-truth images can be particularly difficult or impossible to obtain, e.g., in scientific imaging applications (Belthangady & Royer, 2019), and *(ii)* even when we have access to a ground truth dataset, there might be a significant distribution shift between training and testing.

Self-supervised learning presents an alternative that circumvents some of these limitations (Belthangady & Royer, 2019). This approach operates without the need for ground-truth training data, making it especially advantageous in scenarios where obtaining such datasets is expensive or impractical. Furthermore, self-supervised learning can be trained on measurement data directly, thus mitigating the challenge of distribution shift.

Learning from measurement data only is particularly challenging when the forward operator is not invertible (Tachella et al., 2023). If the forward operator is linear, self-supervised learning is possible by assuming that the underlying set of clean signals is invariant to a set of transformations, such as rotations or translations (Chen et al., 2021). This also holds for the non-linear problem of reconstructing heavily quantized signals (1-bit measurements) (Tachella & Jacques, 2023). Here we show that invariance to rotations or translations is not enough for learning from clipped measurement data alone, and instead prove that invariance to amplitude is sufficient for fully self-supervised learning. We propose a self-supervised loss that enforces equivariance to changes in amplitude, and show throughout a series of experiments that our method can achieve performance comparable to fully supervised methods. A preliminary version of this work was presented in Sechaud et al. (2024), and here we generalize to a larger class of signals such as images, and propose a theoretical framework showing sufficient conditions for learning to recover signals from saturated measurements alone. In summary, the contributions of this work are threefold:

- We provide necessary conditions for signal recovery under a known signal set, and also demonstrate, under an invariance assumption, that the signal set can be identified with a sufficient number of observed measurements;

- We propose a self-supervised loss that can be used to train neural networks for audio declipping and High-Dynamic Range (HDR) imaging;

- We show that this method can achieve comparable performance to fully supervised methods on both audio and image data.

The rest of the paper is structured as follows: Section 2 reviews related work concerning self-supervised learning and state-of-the-art declipping methods. Section 3 introduces our theoretical framework, focusing on model identification—i.e., determining underlying signal set from observations—and signal recovery. From there, in Section 4 we describe how one can practically define and minimize a self-supervised cost function with scale invariance. Section 5 presents experimental results that validate our approach.

## 2 Related work

### 2.1 Self-supervised learning for inverse problems

Self-supervised methods allow to train reconstruction networks using only measurement data (Chen et al., 2021). Existing methods can be separated into two categories: the ones that handle the noise of the problem, e.g., Noise2X methods (Lehtinen et al., 2018; Krull et al., 2019), and the others that handle non-invertible forward operators (Tachella et al., 2023). The last set of methods mostly focuses on the case a linear operator where the loss of information is associated to its nullspace. It is possible to overcome this missing information

by *(i)* accessing measurements from several operators with different nullspaces (Daras et al., 2023; Tachella et al., 2023), or *(ii)* by assuming that the signal distribution is invariant to a set of transformations such as translations or rotations, a method known as equivariant imaging (EI) (Chen et al., 2021). The EI framework is well studied for linear inverse problems, both in terms of applications (Scanvic et al., 2024; Chen et al., 2021) and theory (Tachella et al., 2023). To the best of our knowledge, the only case of nonlinear inverse problems analyzed using the invariance approach is one-bit compressed sensing (Tachella & Jacques, 2023).

## 2.2 Audio declipping

Several unsupervised techniques for audio declipping have been proposed (Zaviska et al., 2021; Kitic et al., 2013; Gaultier et al., 2021), many of which make use of variational methods with priors such as sparsity in certain bases (e.g. Fourier domain). Both $\ell_0$ and $\ell_1$-minimization approaches are commonly employed, often solved using dedicated optimization algorithms, with social sparsity as one of the best performing (Siedenburg et al., 2014). Learning-based techniques, such as dictionary learning, have also shown to be effective, where a dictionary is trained from time windows of the clipped signal to represent it as a sparse vector. Additionally, methods incorporating prior knowledge based on human perception (Defraene et al., 2013), or using multichannel data (Ozerov et al., 2016; Gaultier et al., 2018), have been used to improve performance. Diffusion models trained on a supervised way (i.e., with ground-truth references) have also been studied in the context of declipping (Moliner et al., 2023).

## 2.3 High Dynamic Range images

The dynamic range in images refers to the difference between the brightest and darkest values. Since camera sensors and displays have limitations in capturing and showing both dark and bright regions, this often results in lost details in these zones. To address this, various techniques are employed to extend the dynamic range of images. One common method is exposure bracketing (Mertens et al., 2007), which combines multiple photos at different exposures for higher quality but requires long exposures and a stationary target to avoid misalignment. Another approach is modulo sensing (Contreras et al., 2024), a promising technique, though it is still limited by the unavailability of commercial hardware. Finally, supervised deep learning methods, such as the ones proposed by Eilertsen et al. (2017), utilize a single image to estimate a high dynamic range. However, these methods require a large dataset of ground truths to train the model effectively, which can be a significant limitation. For example, in ultra-high-speed imaging (thousands of frames per second), sensors must trade off dynamic range for acquisition speed. As a result, high-contrast and transient phenomena—such as combustion in rocket launch site monitoring—remain challenging to capture accurately, even with the most high-end cameras (Giassi et al., 2015; Tao et al., 2025).

## 2.4 Signal recovery guarantees for saturated measurements

Some theoretical works have analysed the feasibility of the signal recovery problem from saturated observations. Following a compressive sensing approach (Candes & Wakin, 2008), Foucart & Needham (2016) present unique recovery guarantees for the case of Gaussian measurements and using an $\ell_1$ minimization approach, showing that sparse vectors with low magnitude can be uniquely recovered with high probability.

# 3 Analysis for model identification and signal recovery

## 3.1 Problem formulation

We consider the forward formulation $\boldsymbol{y} = \eta(\boldsymbol{x})$, where $\eta : \mathbb{R} \to \mathbb{R}$ is the element-wise clipping operator with threshold levels $\mu_1, \mu_2$:

$$\eta(u) = \begin{cases} \mu_1 & \text{if} \quad u \leq \mu_1, \\ u & \text{if} \quad u \in (\mu_1, \mu_2), \\ \mu_2 & \text{if} \quad u \geq \mu_2. \end{cases}$$

For our examples with audio and synthetic signals, we use a symmetric threshold $\mu_1 = -\mu_2 = \mu$. For images, we only consider an upper threshold, where $\mu_1 = 0$ and $\mu_2 = 1$, assuming positive signals $\boldsymbol{x}$, as camera sensors

do not suffer saturation from below. Indeed, under low-light conditions, the sensor produces a weaker signal, which may lead to noise or reduced image quality, but not to saturation. We assume that $\mu_1$ and $\mu_2$ are known and that we have a dataset of saturated signals $\{\boldsymbol{y}_i\}_{i=1}^{N}$. We aim to learn the reconstruction function $f_{\boldsymbol{\theta}}$ using the measurement dataset alone. We will therefore consider the following two theoretical questions: *(i) model identification*: for a forward operator $\eta$ and measurement sets $\mathcal{Y}$, does a unique signal set $\mathcal{X}$ exist satisfying some mild priors and $\mathcal{Y} = \eta(\mathcal{X})$? In other words, we want to know if it is possible to find the support of the signal distribution $\mathcal{X}$ from measurement data $\{\boldsymbol{y}_i\}_{i=1}^{N}$ alone as $N \to \infty$, and some mild priors on $\mathcal{X}$ (e.g. invariance to transformations). If the set $\mathcal{X}$ can be identified, we could then hope to recover the signal $\boldsymbol{x}$ from a single observation $\boldsymbol{y}$ via the following program:

$$\hat{\boldsymbol{x}} \in \arg\min_{\boldsymbol{x} \in \mathcal{X}} \|h(\boldsymbol{x}) - \boldsymbol{y}\|^2. \tag{3}$$

*(ii) Signal recovery*: is there a unique solution to the problem in (3)? In other words, can we uniquely recover the signal $\boldsymbol{x}$ from observations $\boldsymbol{y}$ knowing the set $\mathcal{X}$? There may be a unique solution for one, both, or neither of the problems (Tachella et al., 2023). Blind compressed sensing is an example where we can have signal recovery if the dictionary is known, but we cannot identify the dictionary from measurement data alone (Gleichman & Eldar, 2011). On the contrary, we can identify the signal set from rank-one measurements without being able to recover the signals associated with each measurement (Chen et al., 2015). Solving these two theoretical problems allows us to assess the feasibility of learning to reconstruct from measurement data alone, using the self-supervised loss proposed in Section 4. In the following, we will use "identify" only for finding the set $\mathcal{X}$ given a measurement set $\mathcal{Y}$, and "recovery" for finding a vector $\boldsymbol{x}$ given a measurement $\boldsymbol{y}$.

## 3.2 Model identification.

In this section, we will focus on the case where the full set of measurements $\mathcal{Y}$ is observed, although in practice we only observe $\{\boldsymbol{y}_i\}_{i=1}^{N}$. We thus study the conditions for identifying $\mathcal{X}$ from the set of saturated signals $\mathcal{Y} = \eta(\mathcal{X})$.

It may be interesting to note that in some situations we can easily identify the signal set. This happens in the trivial case where the whole signal set is below the saturation threshold.

**Example 1.** *In the symmetric case with threshold level $\mu$ and no additional priors on $\mathcal{X}$, if $\|\boldsymbol{y}\|_\infty < \mu$ for all $\boldsymbol{y} \in \mathcal{Y}$, then $\mathcal{X} = \mathcal{Y}$.*

However, in general without any assumptions on $\mathcal{X}$ neither $\mathcal{Y}$, we cannot hope to guarantee identification, as shown in Remark 1. We thus need to consider some assumptions on $\mathcal{X}$ to identify the signal set from $\mathcal{Y}$. As done in equivariant imaging (Chen et al., 2021), we can consider the group of transformations $G$ under which the signal set is invariant. The reason is that the invariance to transformations $\{\boldsymbol{T}_g\}_{g \in G}$ gives us access to the sets $\mathcal{Y}_g = \eta(\boldsymbol{T}_g \mathcal{X})$ for all $g \in G$. Indeed, we can see $\mathcal{Y}$ as a set of measurements associated with the set $\mathcal{X}$ for different forward operators $\eta(\boldsymbol{T}_g \cdot)$, as we have that:

$$\mathcal{Y}_g = \eta(\boldsymbol{T}_g \mathcal{X}) = \eta(\mathcal{X}) = \mathcal{Y} \quad \text{if} \quad \boldsymbol{T}_g \mathcal{X} = \mathcal{X}.$$

We then need to identify which groups of transformations can help in identifying $\mathcal{X}$. We cannot use the classical groups considered in the linear framework (shift or rotation) as these transformations commute with the operator $\eta$. As shown in the following proposition, if the transformations commute with $\eta$, they do not help to identify the signal set.

**Proposition 1.** *Let $G$ be a group, if we cannot identify $\mathcal{X}$ from $\mathcal{Y} = \eta(\mathcal{X})$ and if for all $g \in G$, the transformations $\boldsymbol{T}_g$ commute with $\eta(\cdot)$, then we cannot identify $\mathcal{X}$ from $\mathcal{Y}_g = \eta(\boldsymbol{T}_g \mathcal{X})$ for all $g \in G$.*

*Proof.* Let $G$ such that for all $g \in G$, $\boldsymbol{T}_g \eta(\cdot) = \eta(\boldsymbol{T}_g \cdot)$. We have access to all measurements associated with the different transformations of the signals:

$$\bigcup_{g \in G} \mathcal{Y}_g = \{y_g = \eta(T_g \boldsymbol{x}) : \forall g \in G, \ \forall \boldsymbol{x} \in \mathcal{X}\}.$$

As $\boldsymbol{T}_g$ and $\eta$ commute, we have

$$\bigcup_{g \in G} \mathcal{Y}_g = \{y_g = \boldsymbol{T}_g \eta(\boldsymbol{x}) : \forall g \in G, \ \forall \boldsymbol{x} \in \mathcal{X}\} = \bigcup_{g \in G} \boldsymbol{T}_g \eta(\mathcal{X}),$$

and so we cannot identify $\mathcal{X}$ from that set since it doesn't bring us more information than $\eta(\mathcal{X})$; the set of measurements is a function of $\eta(\mathcal{X})$. $\qquad \square$

Example 1 shows us that the norm of $\boldsymbol{x}$ has an impact on the feasibility of reconstructing the signal, which gets us to consider the group action of scaling.

**Definition 1** (Scale invariance). *We say that $\mathcal{X}$ is scale invariant if*

$$g\mathcal{X} = \mathcal{X} \text{ for all } g \in \mathbb{R}_+^*. \tag{4}$$

*A set that is scale-invariant is called a cone (Boyd & Vandenberghe, 2004).*

**Proposition 2.** *For any two distinct conic sets $\mathcal{X} \neq \mathcal{X}'$, we have that $\eta(\mathcal{X}) \neq \eta(\mathcal{X}')$.*

*Proof.* Let the open set $\mathbb{B}_\mu = \{\boldsymbol{u} \in \mathbb{R}^n \ : \ \|\boldsymbol{u}\|_\infty < \mu\}$ with radius $\mu > 0$. We recall that we note $\mathcal{Y}$ the set of measurement $\mathcal{Y} = \eta(\mathcal{X})$. Taking the set of unsaturated signals in $\mathcal{X}$, that is $\mathcal{X} \cap \mathbb{B}_\mu$, the operator $\eta$ has no impact on them and therefore

$$\mathcal{X} \cap \mathbb{B}_\mu = \eta(\mathcal{X} \cap \mathbb{B}_\mu).$$

Moreover, if $\boldsymbol{x}$ is in $\mathbb{B}_\mu$, then $|\eta(\boldsymbol{x})| < \mu$, therefore

$$\eta(\mathcal{X} \cap \mathbb{B}_\mu) \subset \mathcal{Y} \cap \mathbb{B}_\mu.$$

But if $\boldsymbol{y}$ is in $\eta(\mathcal{X}) \cap \mathbb{B}_\mu$, then $\boldsymbol{y}$ never reaches the threshold $\mu$ or $-\mu$, and there is thus an $\boldsymbol{x}$ in $\mathcal{X} \cap \mathbb{B}_\mu$ such that $\boldsymbol{y} = \eta(\boldsymbol{x}) = \boldsymbol{x}$, i.e.,

$$(\eta(\mathcal{X}) \cap \mathbb{B}_\mu) \subset (\mathcal{X} \cap \mathbb{B}_\mu)$$

showing finally that

$$\mathcal{Y} \cap \mathbb{B}_\mu = \mathcal{X} \cap \mathbb{B}_\mu.$$

Defining $\hat{\mathcal{X}} = \{g\boldsymbol{y} \ : \ g \in \mathbb{R}_+^*, \ \boldsymbol{y} \in \mathcal{Y} \cap \mathbb{B}_\mu\}$ as the conic extension of $\mathcal{Y} \cap \mathbb{B}_\mu$. We will show that $\hat{\mathcal{X}} = \mathcal{X}$.

**Inclusion $\mathcal{X} \subset \hat{\mathcal{X}}$ :** Let $\boldsymbol{x} \in \mathcal{X}$. Then, there exists $c > 0$ such that $\boldsymbol{y} = c\boldsymbol{x}$ belongs to $\mathcal{X} \cap \mathbb{B}_\mu$. Therefore, $\boldsymbol{x} = \frac{1}{c}\boldsymbol{y}$ belongs to $\hat{\mathcal{X}}$, which implies $\mathcal{X} \subset \hat{\mathcal{X}}$.

**Inclusion $\hat{\mathcal{X}} \subset \mathcal{X}$ :** Let $\hat{\boldsymbol{x}} \in \hat{\mathcal{X}}$. By definition, there exist $g \in \mathbb{R}_+^*$ and $\boldsymbol{y} \in \mathcal{Y} \cap \mathbb{B}_\mu$ such that $\hat{\boldsymbol{x}} = g\boldsymbol{y}$. Since $\mathcal{Y} \cap \mathbb{B}_\mu = \mathcal{X} \cap \mathbb{B}_\mu$, we have $\boldsymbol{y} \in \mathcal{X}$. As $\mathcal{X}$ is a cone, it follows that $\hat{\boldsymbol{x}} = g\boldsymbol{y} \in \mathcal{X}$. Thus, we conclude that $\hat{\mathcal{X}} \subset \mathcal{X}$, completing the proof.

$\qquad \square$

**Remark 1.** *For asymmetric clipping, the same result can be obtained for $\mu_1 \leq 0 < \mu_2$ and $\mathcal{X} \subset \mathbb{R}_+^n$. However, we can find a 2-dimensional counter example for $0 < \mu_1 < \mu_2$: we choose $\mathcal{X}_1 = \{a\boldsymbol{x}_1 \ : \ a \in \mathbb{R}^+\}$ and $\mathcal{X}_2 = \{a\boldsymbol{x}_2 \ : \ a \in \mathbb{R}^+\}$ with $\boldsymbol{x}_1 = (\mu_2, \frac{\mu_1}{2})^\top$ and $\boldsymbol{x}_2 = (\mu_2, \frac{\mu_1}{3})^\top$. Then $\mathcal{X}_1$ and $\mathcal{X}_2$ have the same measurement set:*

$$\mathcal{Y} = \{(t, \mu_1) : t \in (\mu_1, \mu_2)\} \cup \{(\mu_2, t) : t \in (mu_1, \mu_2)\}.$$

## 3.3 Signal recovery

Sufficient conditions for signal recovery generally depend on the dimension of $\mathcal{X}$ (Sauer et al., 1991). Following previous work in compressed sensing (Sauer et al., 1991; Falconer, 2013), we use the box-counting dimension, a measure of the complexity of a set which generalizes several notions of dimension.

**Definition 2.** *The upper box-counting dimension of a compact subset $\mathcal{X}$ is*

$$\dim(\mathcal{X}) = \limsup_{\epsilon \to 0} \frac{\log\left[N_{\mathcal{X}}(\epsilon)\right]}{-\log(\epsilon)},$$

*where $N_{\mathcal{X}}(\epsilon)$ is the covering number, i.e., the minimum number of closed balls of radius $\epsilon$ (with respect to the norm $\|\cdot\|_2$) with centres in $\mathcal{X}$ needed to cover $\mathcal{X}$.*

It can be applied to various structures:

- Vector spaces: The box-counting dimension of a space $\mathbb{R}^k$ intersected with the unit ball is $k$, corresponding to the classical geometrical dimension.

- Manifolds: For a manifold of dimension $k$, the box-counting dimension is $k$, consistent with its geometric dimension.

- Bounded $k$-sparse sets: In high dimensional spaces, the box-counting dimension reflects the effective dimension $k$ rather than the ambient dimension $n$.

Thus, the box-counting dimension uniquely determines the dimension of these various objects, providing a flexible framework for measuring the dimension of objects that can appear in compressive sensing or inverse problems.

We approach the problem of signal recovery by considering, for a conic signal set $\mathcal{X}$, its normalized version:

$$S_{\mathcal{X}} = \left\{ \frac{\boldsymbol{x}}{\|\boldsymbol{x}\|_2} \; : \; \boldsymbol{x} \in \mathcal{X} \right\}$$

and the open bounded set

$$\mathcal{X}_R = \left\{ \boldsymbol{x} \in \mathcal{X} \; : \; \|\boldsymbol{x}\|_2 < R \right\}.$$

When studying the signal recovery problem, one observes that pathological cases where $\eta(\boldsymbol{x}_1) = \eta(\boldsymbol{x}_2)$ for $\boldsymbol{x}_1 \neq \boldsymbol{x}_2$ may arise even when $\dim(\mathcal{X})$ is small, as illustrated in Figure 1. As the operator $\eta$ acts as a projection onto the cube $\mathbb{B}_\mu$ if the signal set is orthogonal to one of the cube's faces, it becomes impossible to recover a signal once it reaches that face. However, we expect such cases to be relatively rare, and therefore we study the problem by randomizing the orientation. To this end, as commonly employed in signal recovery analyses (Ahmed et al., 2013), we consider the operator $\eta(\boldsymbol{A}\cdot)$, where $\boldsymbol{A} \in \mathbb{R}^{m \times n}$ is a random Gaussian matrix, which acts approximaly as a random rotation of the signal set. An illustration is provided in Figure 1. The following theorem provides sufficient conditions on the number of measurements and the maximum radius $R$ to ensure, with high probability, that distances between points in the set $\mathcal{X}_R$ are approximately preserved after applying $\eta(\boldsymbol{A}\cdot)$. Section 5.2.1 provides a simplified example demonstrating how this matrix effectively addresses degenerate cases.

**Theorem 1.** *Let us assume that the normalized set $S_{\mathcal{X}}$ has a finite upper box-counting dimension $\dim(S_{\mathcal{X}}) < k$, such that $N_{\mathcal{X}}(\epsilon) \leq \epsilon^{-k}$ for all $\epsilon < \epsilon^*$ with $\epsilon^* \in (0, 1/2)$. For $m, n > 0$, given a random matrix $\boldsymbol{A} \in \mathbb{R}^{m \times n}$ with i.i.d. entries $A_{ij} \sim \mathcal{N}(0,1)$, a threshold $\mu > 0$, and a radius $R > 0$, there exists an absolute constant $C > 0$ such that, if*

$$m \geq k \log(1/\epsilon^*) \ \text{and} \ R < \mu(\tfrac{1}{2} - \tfrac{k+1}{m}),$$

*then*

$$\mathbb{P}\left(\eta(\boldsymbol{A}\boldsymbol{x}) = \eta(\boldsymbol{A}\boldsymbol{u}) \ \text{for some} \ \boldsymbol{x} \neq \boldsymbol{u} \in \mathcal{X}_R\right) \leq 12e^{-Cm}. \tag{5}$$

*In other words, we measure the probability that the mapping $\eta \circ \boldsymbol{A}$ is not injective (and therefore impossible to invert) over $\mathcal{X}_R$.*

Theorem 1 shows that the event of picking two vectors of $\mathcal{X}_R$ with identical image through $\eta \circ \boldsymbol{A}$ has a probability that decays exponentially fast with the dimension $m$ if this dimension is large compared to the

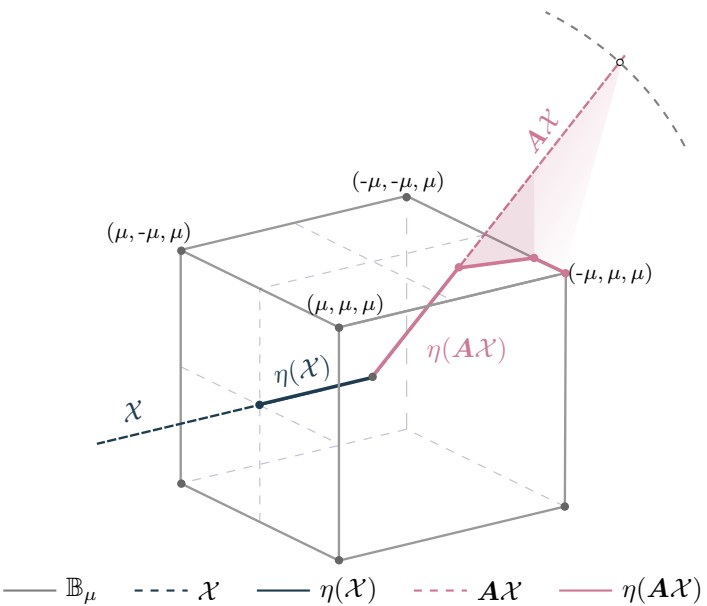

Figure 1: Example in 3 dimensions illustrating Theorem 1. The black set presents a particularly challenging scenario in which all saturated signals are projected to the same point. In contrast, with high probability on $\boldsymbol{A}$, the colored set enables recovery of more signals with moderate norms, as points in $\mathcal{X}$ beyond a certain radius— which is connected to the signal norm when $\boldsymbol{A}$ is Gaussian—are all projected onto a single corner, showing the non-injectivity beyond this radius.

dimension $k$ of the normalized set $S_{\mathcal{X}}$, and the radius is not bigger than $\mu(\frac{1}{2} - \frac{k+1}{m})$. This is quite an interesting effect considering that that the fraction of components of $\boldsymbol{Ax}$ or $\boldsymbol{Au}$ that are impacted by the saturation $\eta$ is constant (in average) when $m$ increases. Indeed, for $\boldsymbol{x} \in \mathcal{X}_R$, each entry of $\boldsymbol{Ax}$ is distributed as a Gaussian random variable with zero mean and standard deviation $\|\boldsymbol{x}\|$. Therefore, for $g \sim \mathcal{N}(0,1)$ and if $m$ is large enough, the fraction of saturated projections $m^{-1}|\{1 \leq i \leq m : |(\boldsymbol{Ax})_i| \geq \mu\}|$ is close to $m^{-1}\sum_{i=1}^{m} \mathbb{P}(|(\boldsymbol{Ax})_i| \geq \mu) = \mathbb{P}(|g|\|\boldsymbol{x}\| \geq \mu)$, which does not depend on $m$.

To consider the initial problem, that is, the forward operator $\eta(\cdot)$, we may introduce an assumption on $\mathcal{X}$ by considering it as a cone of the form

$$\mathcal{X} = \{\boldsymbol{Az} : \boldsymbol{z} \in \mathcal{Z}\},$$

where $\mathcal{Z}$ is a cone with $S_{\mathcal{Z}}$ of box-dimension $k$ and $\boldsymbol{A} \in \mathbb{R}^{n \times n}$ is such that $A_{ij} \sim \mathcal{N}(0,1)$. We can apply Theorem 1 in the case of a random rotation $m = n$, to show that a signal $\boldsymbol{x}$ can be recovered with high probability if $\|\boldsymbol{x}\|_2 < \|\boldsymbol{A}\|R$, where $\|\boldsymbol{A}\|$ denotes the norm of the operator $\boldsymbol{A}$, and in the case of Gaussian square matrix of variance 1, it is close to $2\sqrt{n}$ (Vershynin, 2018).

The proof of the theorem is provided in the Section A, and sketched in Figure 1. It consists in showing that the same measurement vector cannot be associated with two distinct signals, with high probability. The core idea of the proof is to decompose the probability that the mapping is not injective into two parts: Given two distinct signals, we control, the probability that the number of common unsaturated measurements is small (thus making it impossible to differentiate between 2 signals). On the other hand, we control the probability that two signals have the same common unsaturated measurements given that the number of saturated measurements is low and so it is acceptable to ignore them.

Let us stress that in Theorem 1 we cannot set an upper bound on the probability in (5) for all signals in $\mathcal{X}$ and need to limit this space to bounded signals from $\mathcal{X}_R$. Some signals in $\mathcal{X}$ with an excessive norm cannot be recovered as they have too numerous saturated measurements. In Figure 1, this is illustrated by the accumulation, after saturation, of signals in the corners of the cube, particularly when signals have high norms, i.e., far from the cube's center. When all measurements are saturated, or equivalently when $\mu$ tends to 0, we expect to recover the behavior of 1-bit compressive sensing (Tachella & Jacques, 2023), where $\boldsymbol{x}$

can only be estimated up to a non-zero error. This is why we focus on signals with low norms. Theorem 1 can be seen as a generalization of Foucart & Needham (2016, Theorem 1), to more general low-dimensional sets, going beyond sets of $k$-sparse signals.

## 4 Self-supervised learning approach

Given the positive theoretical results regarding model identification and signal recovery of the previous section, we now propose a method for learning from saturated signals $\{\boldsymbol{y}_i = \eta(\boldsymbol{x}_i)\}_{i=1}^N$. We first introduce the loss functions used to train the network, then describe the network architecture and implementation details used for the reconstruction.

### 4.1 Loss functions

**Measurement consistency (MC) loss:** in the absence of noise, a good reconstruction method should provide an estimate that is consistent with the observations, i.e., for all measurements $\boldsymbol{y}$, we should have

$$\eta(f_{\boldsymbol{\theta}}(\boldsymbol{y})) = \boldsymbol{y}.$$

Measurement consistency means that the reconstructed signal produces the same measurements as those that were originally observed. Ensuring MC is crucial, as it guarantees that estimates adhere to observations. However, it does not necessarily mean that the solution is unique since multiple solutions could be consistent with the same measurements. A straightforward loss function for imposing measurement consistency is

$$\mathcal{L}_{\mathrm{NMC}}(\boldsymbol{\theta}) = \sum_{i=1}^N \|\boldsymbol{y}_i - \eta\left(f_{\boldsymbol{\theta}}(\boldsymbol{y}_i)\right)\|^2, \tag{6}$$

which we will call the naive measurement consistency loss. If at the first training steps, the network predicts a value $\hat{x}_j$ above the threshold, the gradient will be zero due to the $\eta$ operator and thus get stuck on this prediction, preventing the network from learning. This is particularly problematic if $y_j$ is below the threshold, since we know that the true value is $x_j = y_j$. We therefore choose to use the MSE where the signal is not saturated according to the following loss:

$$\mathcal{L}_{\mathrm{MC}}(\boldsymbol{\theta}) = \sum_{i=1}^N \|\rho(f_{\boldsymbol{\theta}}(\boldsymbol{y}_i), \boldsymbol{y}_i)\|^2 \tag{7}$$

where $\rho$ is applied element-wise as:

$$\rho(a, b) = \mathbb{1}_{\{|b| < \mu\}}|b - a| + \mathbb{1}_{\{b=\mu\}}(b - a)_+ + \mathbb{1}_{\{b=-\mu\}}(a - b)_+,$$

where $\mathbb{1}(\cdot)$ the indicator function and $(\cdot)_+$ the positive function. See Figure 2 for a comparison of these functions.

However, even with this new loss, we cannot hope to learn the correct reconstruction function. Minimizing $\mathcal{L}_{\mathrm{MC}}$ allows us to learn a measurement consistent solution $f_{\boldsymbol{\theta}}$, but this solution could be of the form:

$$f_{\boldsymbol{\theta}}(\boldsymbol{y})_j = \begin{cases} y_j & \text{if} \quad |y_j| \leq \mu \\ v(y_j) & \text{if} \quad |y_j| = \mu \end{cases} \tag{8}$$

with $v$ being *any* function lying beyond the threshold level $\mu$ i.e., $v(y_j)$ belongs to the preimage $\eta^{-1}(\{-\mu, \mu\})$. For example, the function $f_{\boldsymbol{\theta}}(\boldsymbol{y}) = \boldsymbol{y}$ is a minimizer of (7).

**Equivariance loss**: To effectively learn in the saturated region, we use the amplitude invariance assumption of the signal set defined above, going beyond the limitations imposed by measurement consistency alone.

To enforce this invariance property on the reconstruction network, we note that since $g\boldsymbol{x} \in \mathcal{X}$, the function $f_{\boldsymbol{\theta}}$ must be capable of reconstructing both $\boldsymbol{x}$ and $g\boldsymbol{x}$ for all $\boldsymbol{x}$ in $\mathcal{X}$ and $g \in \mathbb{R}_+$, that is:

$$f_{\boldsymbol{\theta}}\left(\eta(g\boldsymbol{x})\right) = g\boldsymbol{x} \quad \text{and} \quad f_{\boldsymbol{\theta}}(\eta(\boldsymbol{x})) = \boldsymbol{x}. \tag{9}$$

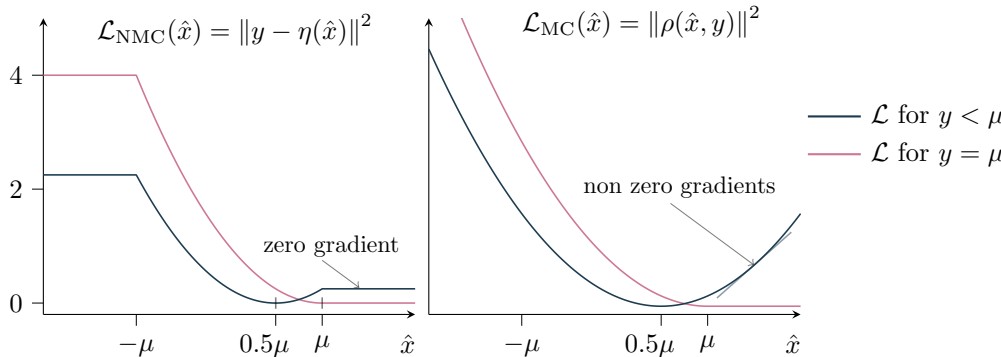

Figure 2: $\mathcal{L}_{\mathrm{NMC}}$ compared to $\mathcal{L}_{\mathrm{MC}}$. On the left: Curve of the $\mathcal{L}_{\mathrm{NMC}}$ in one dimension for two examples in both cases saturated and not. On the right: Curve of the new $\mathcal{L}_{\mathrm{MC}}$ for the same two examples.

Thus, we can conclude that:

$$f_{\boldsymbol{\theta}}\big(\eta(g\boldsymbol{x})\big) = g f_{\boldsymbol{\theta}}\big(\eta(\boldsymbol{x})\big) \text{ for all } \boldsymbol{x} \in \mathcal{X}, \text{ and } g \in \mathbb{R}_+. \tag{10}$$

Hence, the composition of $f_{\boldsymbol{\theta}}$ with $\eta$ should be equivariant with respect to the multiplicative group $(\mathbb{R}_+, \cdot)$. To ensure this, we propose a loss function that enforces this equivariance property:

$$\mathcal{L}_{\mathrm{EI}}(\boldsymbol{\theta}) = \sum_{i=1}^{N} \mathbb{E}_{g \sim p_g} \left[ \left\| g f_{\boldsymbol{\theta}}\left(\boldsymbol{y}_i\right) - f_{\boldsymbol{\theta}}\big(\eta\left(g f_{\boldsymbol{\theta}}\left(\boldsymbol{y}_i\right)\right)\big) \right\|^2 \right]. \tag{11}$$

We can observe a connection with Proposition 2: for $N \to \infty$, after rescaling, the sum becomes an expectation over a distribution supported on $\mathcal{Y}$, and the expectation over $g$ basically acts as a (restricted) conic extension of that support.

The expectation is evaluated with respect to a specific distribution $p_g$ over the multiplicative group, which, in principle, could be any unbounded distribution over $\mathbb{R}_+$. However, in practical scenarios, signals are typically limited in amplitude, making it sufficient to use a distribution defined over a finite range $[g_{\min}, g_{\max}]$ (not necessarily uniform). The parameters $(g_{\min}, g_{\max})$ were determined empirically: setting $g_{\max}$ too low prevents the network from learning effectively, while excessively large values cause instability.

We could also extend (11) to include transformations not limited to scale invariance, such as affine group transformations of the form $\boldsymbol{x} \to a\boldsymbol{x} + b\mathbf{1}$, where $(a, b) \in \mathbb{R}_+ \times \mathbb{R}$ and $\mathbf{1} = (1, \ldots, 1)^{\top} \in \mathbb{R}^n$.

The final loss function is a combination of the measurement consistency and equivariance losses:

$$\mathcal{L}(\boldsymbol{\theta}) = \mathcal{L}_{\mathrm{MC}}(\boldsymbol{\theta}) + \lambda \mathcal{L}_{\mathrm{EI}}(\boldsymbol{\theta}). \tag{12}$$

The parameter $\lambda$ is a positive hyperparameter that controls the trade-off between the two losses.

### 4.2 Network architecture

### 4.2.1 Core architecture

The proposed self-supervised loss can be used with any network architecture $f_{\boldsymbol{\theta}}$. For most of the experiments reported here, we choose the well-known U-Net neural network for the reconstruction function $f_{\boldsymbol{\theta}}$ (Ronneberger et al., 2015), which is well-suited for capturing the relevant (temporal or spatial) correlations in natural signals. The details of the architecture vary depending on the type of data, audio or image, and are explained in the experiment section.

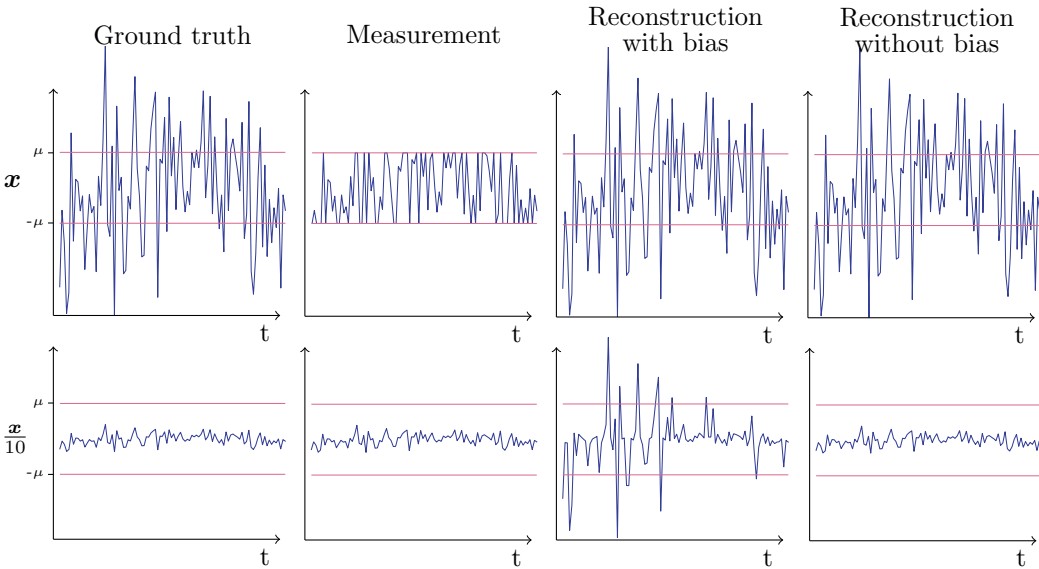

Figure 3: An example where the reconstruction is learned through the bias (third column) which prevents the reconstruction for low amplitude measurements. We thus observe that removing the bias makes the network naturally homogeneous and improves the reconstruction. On the first row, we consider a signal $\boldsymbol{x}$, on the second row $\frac{\boldsymbol{x}}{10}$.

### 4.2.2 Bias-free neural network

We choose a bias-free version of the U-Net, to preserve the scale invariance assumption (Mohan et al., 2020). This is because a feedforward neural network with ReLU activation functions and without bias defines a homogeneous function:

$$f_{\boldsymbol{\theta}}(g\boldsymbol{y}) = g f_{\boldsymbol{\theta}}(\boldsymbol{y}) \quad \forall g \in \mathbb{R}_+.$$

This gives $f_{\boldsymbol{\theta}}(g\boldsymbol{y}) \underset{g \to 0}{\longrightarrow} 0$, which is particularly necessary for unsaturated signals. Figure 3 shows an example where a biased network cannot reconstruct an unsaturated signal.

### 4.3 Masking

A key aspect of the threshold operator is that, by knowing the threshold levels, we can identify the unsaturated, and therefore non-degraded, portion of the signal. While the measurement consistency loss encourages the network to preserve the unsaturated part, it does not guarantee strict adherence to this constraint. To address this limitation, we can enforce measurement consistency more rigorously by employing the following masking method, comparable to a more structured skip connection architecture. The estimated signals $\hat{\boldsymbol{x}}$ are computed using an element-wise linear blending between the outputs of the network $f_{\boldsymbol{\theta}}(\boldsymbol{y})$ and the measurements, that is

$$\hat{x}_j = (1 - b_j)y_j + b_j f_{\boldsymbol{\theta}}(\boldsymbol{y})_j. \tag{13}$$

By choosing an appropriate blend value $b_j$ for all sample points or pixels, the blend works as a mask that modifies only saturated measurements, avoiding artifacts in the unsaturated parts.

$$b_j = \begin{cases} 0 & \text{if } |y_j| < \mu \\ 1 & \text{if } |y_j| = \mu \end{cases}$$

An example of an image and its associated mask is shown in Figure 4.

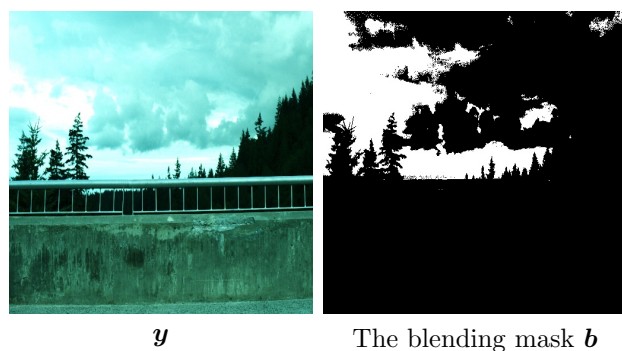

$\boldsymbol{y}$          The blending mask $\boldsymbol{b}$

Figure 4: An image with its associated blending mask. The mask is white where a channel is saturated and black when none is saturated.

## 5 Experiments

We begin by presenting experiments on toy datasets, where we can control the properties of the signal set. The first experiment uses a dataset associated to a one-dimensional vector space that is invariant to amplitude. The second experiment is performed using various datasets generated by sampling vectors from random $k$-dimensional subspaces. To generate these datasets, we first control the dimension of the vector space, followed by the saturation proportion for each sampled vector. The reconstruction is then evaluated based on these varying parameters. Next, we conduct experiments on real-world music data, and we compare our method to several state-of-the-art techniques. Finally, we conclude with experiments on two-dimensional signals, where we aim to reconstruct HDR images.

### 5.1 Metrics

Two common metrics for evaluating the quality of reconstructions are the Signal-to-Distortion Ratio (SDR) and Peak Signal-to-Noise Ratio (PSNR). PSNR is more suitable for image quality assessment, as it incorporates a maximum possible pixel value, allowing for a clearer comparison of signal fidelity. SDR, on the other hand, is generally applied to audio signals because it does not require knowledge of the maximum possible value. Although we do not know this value for HDR images, we still use PSNR to provide a more relevant comparison with other works. The PSNR and SDR metrics are defined as follows:

$$\text{PSNR}(\boldsymbol{x}, \hat{\boldsymbol{x}}) = 20 \log_{10} \left( \frac{1}{\|\boldsymbol{x} - \hat{\boldsymbol{x}}\|_2} \right), \tag{14}$$

$$\text{SDR}(\boldsymbol{x}, \hat{\boldsymbol{x}}) = 20 \log_{10} \left( \frac{\|\boldsymbol{x}\|_2}{\|\boldsymbol{x} - \hat{\boldsymbol{x}}\|_2} \right). \tag{15}$$

Both metrics are referred to as distortion metrics since they assess the fidelity of the reconstruction process. In addition to distortion metrics, perception metrics could be used to evaluate the quality of the reconstruction, offering a more accurate measure of how changes to a signal or image affect human perception. In this work, we employ the widely adopted metric Perceptual Evaluation of Speech Quality (PESQ) to evaluate audio signals. For image, we use the Natural Image Quality Evaluator (NIQE), a no-reference (no ground-truth needed) metric that estimates image quality. While perceptual metrics are particularly relevant for applications like music audio, where subjective experience plays a key role, they are less suitable for signals where preserving critical information is essential, as they tend to permit greater hallucination - introducing elements that were not part of the original signal, potentially compromising the accuracy of the reconstruction. Blau & Michaeli (2018) study the impossibility of algorithms to optimize both perception and distortion metrics, known as the perception-distortion tradeoff.

### 5.2 Toy datasets

#### 5.2.1 MNIST experiment

In this example, we use a toy signal set, generated from a single signal and augmented with its scaled versions, i.e. $\mathcal{X} = \{e\boldsymbol{x}_0 : e \in \mathbb{R}_+^*\}$ for $\boldsymbol{x}_0$ fixed. In practice, we take an image $\boldsymbol{x}$ from MNIST and create the dataset $\mathcal{D} = \{e_i\boldsymbol{x}_0\}_{i=1}^N$ with $e_i$ a realization of a random variable with exponential distribution of mean 2, for $i \in \{1, \dots, N\}$. This dataset has two interesting properties: it is scale invariant, and we cannot have unique signal recovery as shown in Figure 5.

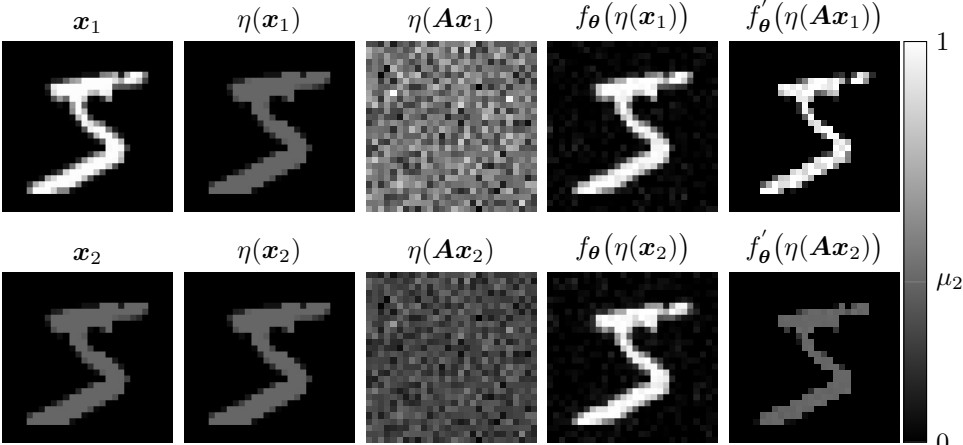

Figure 5: Example where two different signals $\boldsymbol{x}_1, \boldsymbol{x}_2$ have the same measurement $\eta(\boldsymbol{x}_1), \eta(\boldsymbol{x}_2)$ and so the network $f_{\boldsymbol{\theta}}$ fails to recover them both (the second and fourth columns). Adding randomness implies that the two measurements are not equal anymore, and so a network $f_{\boldsymbol{\theta}}'$ can reconstruct the original signal (the third and fifth columns).

Measurements are obtained by applying the forward function $\eta(\boldsymbol{A}\cdot)$ with $\boldsymbol{A}$ an orthogonal matrix and threshold levels $\mu_1 = 0$, $\mu_2 = 0.4$. The matrix $\boldsymbol{A}$ is drawn randomly with respect to the Haar measure (Mezzadri, 2007). While Figure 5 shows that it is impossible without $\boldsymbol{A}$ to recover the original signals, adding randomness in the forward operator solves this issue, as illustrated in Figure 1. Figure 6 shows that the neural network trained using the loss function $\mathcal{L}_{\mathrm{MC}} + \mathcal{L}_{\mathrm{EI}}$ successfully reconstructs images with a dynamic range similar to the original images, whereas the network trained with $\mathcal{L}_{\mathrm{NMC}} + \mathcal{L}_{\mathrm{EI}}$ as the loss function does not achieve this. However, both fail when the dynamic range of the original signal becomes excessively high. For signals with high dynamic range, characterized by a large norm $\|\cdot\|_2$, the network is unable to recover them as indicated in Theorem 1.

#### 5.2.2 Synthetic dataset

First, we generate a random subspace of $\mathbb{R}^{100}$ with dimension $k \in \{1, \dots, 100\}$ using basis vectors whose coordinates are drawn from a standard normal distribution. Next, we generate $N = 1000$ vectors $\{\boldsymbol{x}_i\}_{i=1}^N$ within this subspace, rescaling each vector such that a proportion $v \in (0, 1)$ of their components are clipped with the threshold $\mu$ set to 1.

We evaluate the recovery performance as a function of the parameters $k$ and $v$. Figure 7 illustrates the network's ability to reconstruct unsaturated signals as these parameters vary. The performance decreases as the subspace dimension grows or the proportion of saturated samples increases. Indeed, signals from higher-dimensional subspaces are more challenging to reconstruct, which is consistent with the result of Theorem 1.

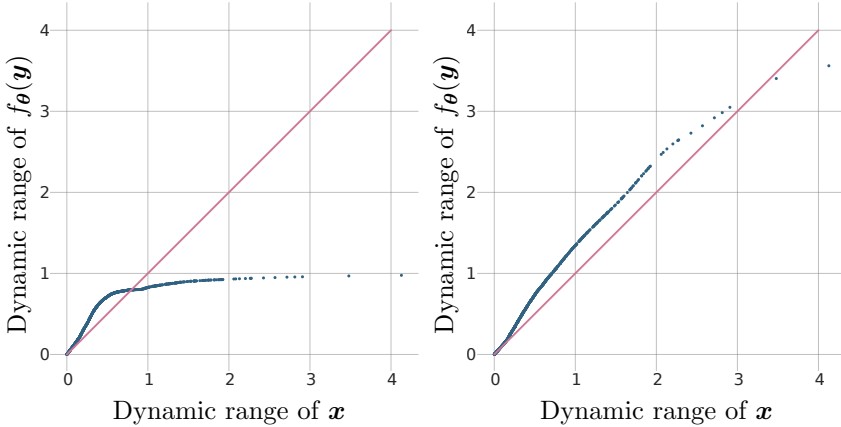

Figure 6: Plot of the dynamic range of $f_{\boldsymbol{\theta}}(\boldsymbol{y})$ depending on dynamic range of $\boldsymbol{x}$, where $\boldsymbol{y} = \eta(\boldsymbol{Ax})$. Left: The training is done with the losses $\mathcal{L}_{\mathrm{NMC}} + \mathcal{L}_{\mathrm{EI}}$. Right: The training is done with loss $\mathcal{L}_{\mathrm{MC}} + \mathcal{L}_{\mathrm{EI}}$. A perfect plot should be the identity line.

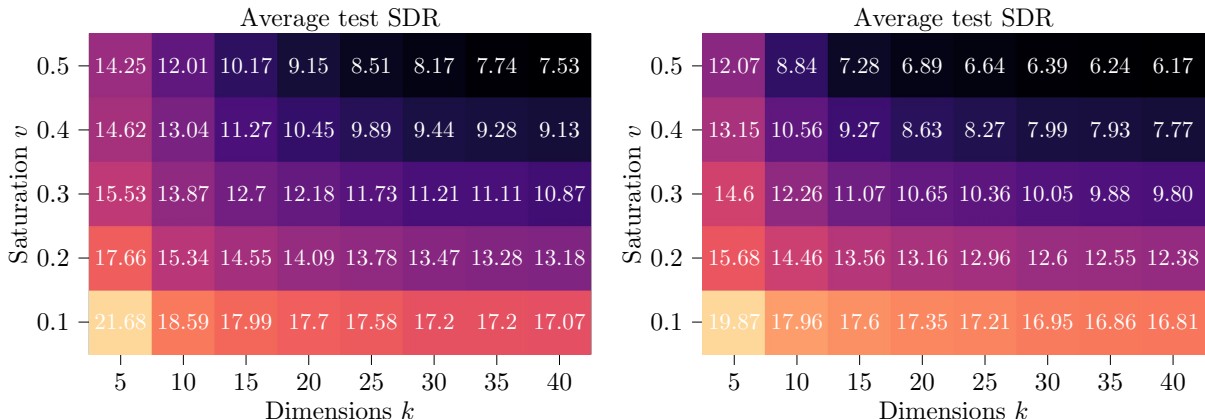

Figure 7: Average reconstruction performance as a function of saturated part $v$ and signal set dimension $k$ for both supervised (left) and self-supervised (right) methods. The value indicated corresponds to the mean SDR over the test dataset.

## 5.3 Audio

We evaluate the method on a real audio dataset. Specifically, we use the GTZAN dataset, which contains 10 genres (rock, classical, jazz, etc.), each represented by 100 audio files, all 30 seconds long with a sampling rate of 22,050 Hz. The audio signals are split into 30 segments of 1-second each and then saturated with a threshold level $\mu = 0.1$. Measurements $\boldsymbol{y}$ without any saturated entries are discarded. This process results in a training dataset of 21898 audio samples and a testing dataset of 100 samples.

The distribution of $p_g$ is set uniformly on the interval $[0.1, 2]$. Results are compared with a state-of-the-art variational method called social sparsity (Zaviska et al., 2021; Siedenburg et al., 2014) and supervised methods. Both the supervised and self-supervised approaches use the same network architecture for training. We refer to the method trained using only 5% of the whole training dataset as "Supervised 5%", which is used to compare the performance of the both approaches under limited ground-truth data acquisition. We also compare our method with the following methods: self-supervised with loss $\mathcal{L}_{\mathrm{MC}}$ alone, supervised with MSE + $\mathcal{L}_{\mathrm{EI}}$. As shown in Table 1, the proposed self-supervised method achieves performance comparable to the fully supervised approach, although the latter requires significantly less training time: 7 hour 46 minutes compared to 13 hours 52 minutes for the self-supervised method. The results further confirm that

training with the MC loss alone does not yield additional information beyond the measurements themselves, as anticipated in Section 4. Similarly, the proposed method with a biased network does not perform as well, illustrating the importance of using a bias-free architecture as seen in Section 4.2.2. In addition, the supervised method with equivariance does not demonstrate any significant improvement over the standard supervised approach.

Table 1: Reconstruction performance, SDR and PESQ are averaged over all music test dataset.

| Methods | SDR ↑ | PESQ ↓ |
|---|---|---|
| Identity | $4.84 \pm 2.12$ | $2.7 \pm 0.65$ |
| Social Sparsity | $9.92 \pm 4.46$ | $2.09 \pm 0.95$ |
| Supervised | $11.69 \pm 2.25$ | $1.94 \pm 0.75$ |
| Supervised 5% | $9.79 \pm 1.59$ | $2.28 \pm 0.78$ |
| Supervised + $\mathcal{L}_{EI}$ | $11.72 \pm 2.23$ | $1.92 \pm 0.74$ |
| Proposed self-supervised | $10.48 \pm 2.20$ | $2.20 \pm 0.80$ |
| Unsupervised with $\mathcal{L}_{MC}$ alone | $4.84 \pm 2.12$ | $2.68 \pm 0.65$ |
| Proposed self-supervised with bias | $9.27 \pm 3.22$ | $2.43 \pm 0.90$ |

To demonstrate that a supervised training dataset may not generalize well to a different test set, we conduct the following experiment: we create a supervised training dataset consisting of only music, while the test dataset includes both music and voice recordings (Vryzas et al., 2018a;b), with only measurement data available (no ground truth). The supervised method, trained using the MSE loss (2), learns solely from the training dataset, whereas the self-supervised method is trained on both the training and test datasets since it does not require ground truth. Both methods are then evaluated on the test dataset. The results, shown in Table 2, demonstrate that the self-supervised method is more robust when the training and test datasets differ.

Table 2: Average SDR performance on the test dataset, which includes both music and voice recordings.

| Methods | SDR (dB) |
|---|---|
| Identity | $6.54 \pm 2.34$ |
| Supervised (trained on music) | $10.94 \pm 2.00$ |
| Proposed self-supervised (trained on music and voice) | $11.92 \pm 2.46$ |

Table 3: Average SDR and PESQ performance on the test MAESTRO dataset.

| Methods | SDR ↑ | PESQ ↓ |
|---|---|---|
| Identity | $5.00 \pm 0$ | $2.9 \pm 0.73$ |
| Supervised | $6.70 \pm 1.69$ | $2.22 \pm 0.52$ |
| Proposed self-supervised | $7.78 \pm 8.64$ | $2.47 \pm 1.12$ |

We also compare the self-supervised learning method to a supervised learning approach using diffusion models (Moliner et al., 2023). The comparison is conducted on a portion of the MAESTRO dataset (Hawthorne et al., 2019), used in Moliner's work, which consists of 200 hours of classical solo piano recordings. The measurement signals from Moliner et al. (2023) are used as a benchmark; although the saturation threshold varies for each signal, it consistently corresponds to an SDR value of 5. Each signal is then normalized to yield a saturation threshold of 0.1, while preserving a constant SDR of 5. This approach enables comparison between the proposed method, based on a fixed saturation threshold, and Moliner's trained method. Results demonstrate that the self-supervised method outperforms the supervised one Table 3. It is important to note that the proposed self-supervised method focuses on optimizing the distortion metric SDR, whereas this supervised method aims to optimize a perceptual metric PESQ. Results are then consistent with the perception-distortion tradeoff seen in Section 5.1.

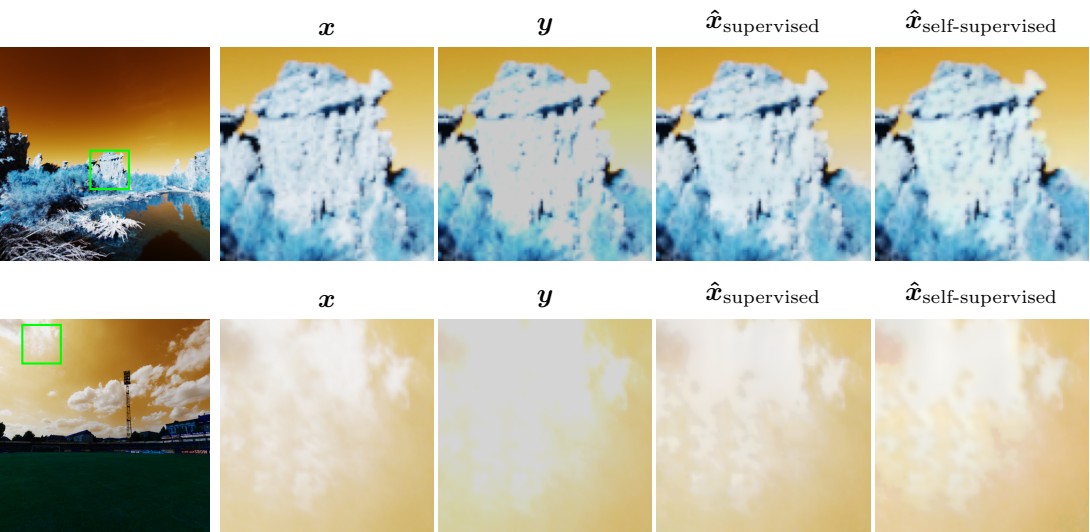

Figure 8: Image reconstruction on the test dataset. An exposition correction (Wikipedia) is applied to see details better.

## 5.4 HDR

The HDR experiment is inspired by Eilertsen et al. (2017), and its goal is to compare supervised and self-supervised methods. Results are summarized in Table 4 and in Figure 8. The dataset is composed of 1043 HDR images noted $\boldsymbol{u}$ (Le et al., 2023), which are considered as real scenes. Photos are taken with a virtual camera calibrated by a camera curve and exposure. For each image, we choose the exposure time to saturate between 5% and 15% of the pixels. We divide the image by the quantile $q_v$ where $v$ is chosen uniformly in $(0.85 , 0.95)$. Then we have $\mathbb{P}\left(\frac{\boldsymbol{u}}{q_v} > 1\right) = 1 - v$ if $\mathbb{P}$ is the cumulative histogram of $\boldsymbol{u}$. The camera curve is defined as

$$\omega(\boldsymbol{u})_j = (1 + \sigma)\frac{u_j^\beta}{u_i^\beta + \sigma},$$

We use these functions to fit the database of real camera curves collected by Grossberg & Nayar (2003). We choose $\beta \sim \mathcal{N}(0.9, 0.1)$ and $\sigma \sim \mathcal{N}(0.6, 0.1)$. The images $\boldsymbol{x} = \omega\left(\frac{\boldsymbol{u}}{q_v}\right)$ represent the ground truth dataset. They are quantized and clipped to create the measurement dataset. This step is performed with:

$$y_j = \frac{\lfloor 255 \min(1, x_j) + 0.5 \rfloor}{255}. \tag{16}$$

We use an unfolded network with a U-Net architecture. It combines traditional iterative optimization with neural network learning. For each iteration, we replace a part of the optimization step with a neural network. We choose the Half-Quadratic Splitting (HQS) (Aggarwal et al., 2018) algorithm which aims to minimize $\varphi(x) + \lambda\psi(x)$, for two closed proper convex functions $\varphi$ and $\psi$. The iteration step is given by:

$$u_k = \text{prox}_{\gamma\varphi}(x_k),$$
$$x_{k+1} = \text{prox}_{\sigma\lambda\psi}(u_k).$$

In our setting, $\varphi$ corresponds to the measurement consistency term $\varphi(\boldsymbol{x}) = \|\rho(\boldsymbol{x}, \boldsymbol{y})\|^2$ and $\text{prox}_{\sigma\lambda\psi}(u_k)$ is replaced by a neural network. The proximal operator of the measurement consistency term is given by:

$$\text{prox}_{\gamma\varphi}(\boldsymbol{x})_j = \begin{cases} \frac{x_j + \gamma y_j}{1+\gamma} & \text{if } |x_j| \le \mu, \\ x_j & \text{otherwise} \end{cases}$$

for $j \in \{1, \ldots, n\}$.

Table 4: Reconstruction performance, PSNR is averaged over all test dataset images.

| Methods | PSNR ↑ | NIQE ↓ |
|---|---|---|
| Identity | $29.54 \pm 6.95$ | $4.21 \pm 1.96$ |
| PnP DPIR | $32.11 \pm 5.80$ | $4.06 \pm 1.93$ |
| Supervised | $36.67 \pm 6.19$ | $4.04 \pm 1.74$ |
| Proposed self-supervised | $35.04 \pm 6.33$ | $4.10 \pm 1.85$ |

In Table 4, we compare our method against two reference approaches: the supervised method from Eilertsen et al. (2017) and the Plug-and-Play (PnP) method (Venkatakrishnan et al., 2013). The PnP method is an iterative technique that integrates a denoising module into the optimization process, replacing the proximal operator. We used the Deep Plug-and-Play Image Restoration (DPIR) method (Zhang et al., 2021), a PnP approach based on HQS in which the denoising network is a pre-trained DRUNet, and the noise level for the denoiser is adapted at each iteration. For better results, we modified the initialization of the algorithm – which is generally set to $\boldsymbol{y}$ – by adding a white Gaussian noise of standard deviation $\sigma = 0.18$, which we found to better avoid local minima close to $\boldsymbol{y}$. In this way, DPIR outperforms the other PnP methods tested. Results in Table 4, demonstrate that the self-supervised method performs on par with the supervised method and significantly outperforms the DPIR approach. As illustrated in Figure 8, both the supervised and self-supervised methods effectively recover details in the images.

## 6 Conclusion

In this study, we propose a self-supervised method that uses amplitude invariance to address the nonlinear declipping problem. We provide a theoretical framework with guarantees for model identification of scale invariant signal models and unique recovery of the unsaturated signals when the operator involves random matrices with coefficients following a Gaussian distribution. Experimental results, carried out on audio and images, indicate that this method can perform on par with the supervised approach, and surpasses variational methods. This work opens up new possibilities for learning-based methods in nonlinear inverse problems.

## 7 Acknowledgment

Victor Sechaud and Julian Tachella are supported by the ANR grant UNLIP (ANR-23-CE23-0013). The reserch of Laurent Jacques is partly funded by FRS-FNRS (QuadSense, T.0160.24). Part of this research was carried out when LJ was supported by the OCKHAM team at ENS Lyon (France), with an INRIA Chair at the Institute for Advanced Study, Collegium of Lyon.

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

# A   Appendix: Proof of Theorem 1

We recall that the idea of the proof is to upper-bound two probabilities: *(i)* the probability that the number of common unsaturated measurements is small; *(ii)* the probability that two signals have the same common unsaturated measurements given that the number of saturated measurements is low. Before we begin, we need to introduce some tools of linear embedding. Theorem 2 allows us to prove Corollary 2.1 used to control the first targeted probability. Theorem 3 is used to control the second one. We also recall the Hoeffding inequality for sub-Gaussian random variables, which we will use to verify an assumption for Theorem 2.

**Lemma 1.  *Hoeffding inequality.*** *Let $X_1, \ldots, X_m$ be independent random variables, where $X_i$ has mean $\mu_i$ and is $\sigma_i$-sub-Gaussian. Then for all $t \geq 0$ we have*

$$\mathbb{P}\bigg( \bigg| \sum_{i=0}^{m} X_i - \mu_i \bigg| \geq t \bigg) \leq 2 \exp\bigg( - \frac{t^2}{2 \sum\limits_{i=0}^{m} \sigma_i^2} \bigg)$$

**Theorem 2.**  *This theorem is adapted from Puy et al. (2017, Theorem II.2).*
*Let us assume that the normalized set $S_{\mathcal{X}}$ has a finite upper box-counting dimension $\dim(S_{\mathcal{X}})$ which is strictly bounded by $k \geq 1$ i.e. $\dim(S_{\mathcal{X}}) < k$, and so there exists a signal set dependent constant $\epsilon^* \in \left(0, \frac{1}{2}\right)$ such that*

$N_{S_\mathcal{X}}(\epsilon) \leq \epsilon^{-k}$ for all $\epsilon \leq \epsilon^*$. Let $\boldsymbol{A} : R^n \to \mathbb{R}^m$ such that $A_{ij} \sim \mathcal{N}\left(0, \frac{1}{m}\right)$. Then there exists $c_1 > 0$ such that, for any $\xi, \delta \in (0,1)$, we have $\sup\limits_{\boldsymbol{x} \in S_\mathcal{X}} |c\|\boldsymbol{Ax}\|_1 - 1| \leq \delta$ with probability at least $1 - \xi$ provided that

$$m \geq \frac{C_1}{c_1 \delta^2} \max\left(k \log\left(\frac{1}{\epsilon^*}\right), \log\left(\frac{6}{\xi}\right)\right),$$

where $C_1$ is an absolute constant.

The proof of Theorem 2 closely follows that of Puy et al. (2017, Theorem II.2). It consists in using a general chaining argument (see, e.g., Vershynin (2018, Chapter 8)) on the function $\boldsymbol{x} \in \mathbb{R}^n \mapsto h(\boldsymbol{x}) := c\|\boldsymbol{Ax}\|_1 - \|\boldsymbol{x}\|_2$ provided that, for any fixed vectors $\boldsymbol{u}, \boldsymbol{v} \in S_\mathcal{X} \cup \{\boldsymbol{0}\}$ and with high probability, the absolute difference $|h(\boldsymbol{u}) - h(\boldsymbol{v})|$ does not deviate too much from the distance $\|\boldsymbol{u} - \boldsymbol{v}\|$, a fact that we prove the following lemma.

**Lemma 2.** *Let $\boldsymbol{A} : \mathbb{R}^n \to \mathbb{R}^m$ such that $A_{ij} \sim \mathcal{N}\left(0, \frac{1}{m}\right)$, $c > 0$, and $h$ be defined as*

$$
\begin{aligned}
h: \quad \mathbb{R}^n &\to \quad \mathbb{R} \\
\boldsymbol{x} &\mapsto \quad c\|\boldsymbol{Ax}\|_1 - \|\boldsymbol{x}\|_2.
\end{aligned}
$$

*Then there exist a constant $c_1 \in \mathbb{R}^+$ such that for any fixed $\boldsymbol{u}, \boldsymbol{v} \in S_\mathcal{X} \cup \{\boldsymbol{0}\}$,*

$$\mathbb{P}\left(|h(\boldsymbol{u}) - h(\boldsymbol{v})| \geq \lambda \|\boldsymbol{u} - \boldsymbol{v}\|\right) \leq 2e^{-c_1 m \lambda^2} \quad \forall \lambda \in \mathbb{R}^+.$$

*Proof.* We consider $h : \boldsymbol{x} \mapsto c\|\boldsymbol{Ax}\|_1 - \|\boldsymbol{x}\|_2$ with $c > 0$. Restricted to $S_\mathcal{X}$, we have $h : \boldsymbol{x} \mapsto c\|\boldsymbol{Ax}\|_1 - 1$.

$$\mathbb{P}\left(|h(\boldsymbol{u}) - h(\boldsymbol{v})| \geq \lambda \|\boldsymbol{u} - \boldsymbol{v}\|\right) \tag{17}$$

$$= \mathbb{P}(c|\|\boldsymbol{Au}\|_1 - \|\boldsymbol{Av}\|_1| \geq \lambda \|\boldsymbol{u} - \boldsymbol{v}\|)$$

$$= \mathbb{P}\left(c\left|\sum_{i=0}^m |\boldsymbol{a}_i^\top \boldsymbol{u}| - |\boldsymbol{a}_i^\top \boldsymbol{v}|\right| \geq \lambda \|\boldsymbol{u} - \boldsymbol{v}\|\right). \tag{18}$$

We note $X_i = |\boldsymbol{a}_i^\top \boldsymbol{u}| - |\boldsymbol{a}_i^\top \boldsymbol{v}|$. Let's show that $X_i$ is sub-Gaussian: For all $q \in \mathbb{N}$ and for all $i \in \{1, \ldots, m\}$,

$$\mathbb{E}\left[|X_i|^q\right] = E\left[\left||\boldsymbol{a}_i^\top \boldsymbol{u}| - |\boldsymbol{a}_i^\top \boldsymbol{v}|\right|^q\right] \leq E[|\boldsymbol{a}_i^\top (\boldsymbol{u} - \boldsymbol{v})|^q]$$

by the reverse triangle inequality. As

$$\boldsymbol{a}_i^\top (\boldsymbol{u} - \boldsymbol{v}) \sim \mathcal{N}\left(0, \frac{\|\boldsymbol{u} - \boldsymbol{v}\|_2^2}{m}\right),$$

$\boldsymbol{a}_i^\top (\boldsymbol{u} - \boldsymbol{v})$ is Gaussian and thus $X_i$ is also sub-Gaussian with parameter $\sigma_i^2 \leq \frac{\|\boldsymbol{u}-\boldsymbol{v}\|_2^2}{m}$. We can therefore apply the Lemma 1:

$$\mathbb{P}\left(c\left|\sum_{i=0}^m X_i\right| > t\right) \leq 2\exp\left(-\frac{t^2}{2c^2 \sum\limits_{i=0}^m \sigma_i^2}\right)$$

and thus give an upper bound of (18):

$$\mathbb{P}(|h(\boldsymbol{u}) - h(\boldsymbol{v})| \geq \lambda \|\boldsymbol{u} - \boldsymbol{v}\|)$$

$$\leq 2\exp\left(-\frac{(\lambda\|\boldsymbol{u} - \boldsymbol{v}\|)^2}{2c^2 \sum\limits_{i=0}^m \sigma_i^2}\right)$$

$$\leq 2\exp\left(-\frac{(\lambda\|\boldsymbol{u} - \boldsymbol{v}\|)^2}{2c^2 \sum\limits_{i=0}^m \frac{\|\boldsymbol{u}-\boldsymbol{v}\|_2^2}{m}}\right)$$

$$\leq 2\exp\left(-\frac{\lambda^2}{2c^2}\right)$$

$$= 2\exp\left(-mc_1 \lambda^2\right).$$

We conclude the proof with $c_1 = \dfrac{1}{2mc^2}$. $\qquad\square$

**Corollary 2.1.** *Under assumptions of Theorem 2, we have for all $m \geq k\log\left(\dfrac{1}{\epsilon^*}\right)$:*

$$\mathbb{P}\big(\forall \boldsymbol{x} \in \mathcal{X} : \|\boldsymbol{A}\boldsymbol{x}\|_1 \leq \sqrt{m}\|\boldsymbol{x}\|_2\big) > 1 - 6e^{-Cm}.$$

*Proof.* By applying Theorem 2 for $c = \frac{2}{\sqrt{m}}$ we deduce that with $c_1 = \frac{1}{2mc^2} = \frac{1}{8}$, for any $\xi, \delta \in (0,1)$, we have with probability $1 - \xi$,

$$\forall \boldsymbol{z} \in S_{\mathcal{X}}, \quad \left|\frac{2}{\sqrt{m}}\|\boldsymbol{A}\boldsymbol{z}\|_1 - 1\right| \leq \delta \tag{19}$$

as long as

$$m \geq \frac{C_1}{c_1 \delta^2} \max\left(k\log\left(\frac{1}{\epsilon^*}\right), \log\left(\frac{6}{\xi}\right)\right).$$

For all $\boldsymbol{x} \in \mathcal{X}$, we obtain by replacing $\boldsymbol{z}$ by $\frac{\boldsymbol{x}}{\|\boldsymbol{x}\|}$ in (19),

$$\|\boldsymbol{A}\boldsymbol{x}\|_1 \leq (\delta + 1)\frac{\sqrt{m}}{2}\|\boldsymbol{x}\|_2 \tag{20}$$

provided that

$$m \geq \frac{8C_1}{\delta^2} \max\left(k\log\left(\frac{1}{\epsilon^*}\right), \log\left(\frac{6}{\xi}\right)\right).$$

By setting $C = \frac{1}{8C_1}$, $\delta = 1$ and $\xi = 6\exp\big(-Cm\big)$, we can verify that the last inequality holds for all $m \geq k\log\left(\dfrac{1}{\epsilon^*}\right)$ and thus:

$$\mathbb{P}\big(\forall \boldsymbol{x} \in \mathcal{X} : \|\boldsymbol{A}\boldsymbol{x}\|_1 \leq \sqrt{m}\|\boldsymbol{x}\|_2\big) > 1 - 6e^{-Cm}.$$

$\qquad\square$

**Theorem 3.** *Adapted from Robinson (2010, Thm. 4.3) Let a compact set $\mathcal{X}$ with finite upper box-counting dimension $\dim(\mathcal{X}) < k$ and $\boldsymbol{A} \in \mathbb{R}^{m \times n}$ such that $A_{ij} \sim \mathcal{N}\left(0, \frac{1}{m}\right)$. If $m > 2k$ then*

$$\mathbb{P}\left(\boldsymbol{A}\boldsymbol{x} = \boldsymbol{A}\boldsymbol{u} \text{ for some } \boldsymbol{x} \neq \boldsymbol{u} \in \mathcal{X}\right) = 0$$

Note that Robinson's theorem states that almost every linear map $\boldsymbol{L}\colon \mathbb{R}^N \to \mathbb{R}^k$ (for $m > 2k$) embeds $\mathcal{X}$ injectively with Hölder distortion $\|\boldsymbol{x} - \boldsymbol{y}\| \leq C\|\boldsymbol{L}\boldsymbol{x} - \boldsymbol{L}\boldsymbol{y}\|^\rho$, for some $\rho \in (0,1)$. By considering the definition of "almost every" in the sense of measure, we can say that the set of non-injective linear maps has zero measure, and therefore zero probability.

For the rest of the proof, we will consider

$$\mathcal{X}_R = \mathcal{X} \cap \mathbb{B}_{\|.\|_2}\big(0, \sqrt{m}\mu\alpha\big)$$

with $R = \sqrt{m}\mu\alpha$ and $\alpha > 0$ to be define latter. We define

$$\mathcal{I}_{\text{sat}}(\boldsymbol{y}) = \{j : |y_j| \geq \mu\}$$
$$\overline{\mathcal{I}_{\text{sat}}}(\boldsymbol{y}) = \{j : |y_j| < \mu\}$$

as the sets of the index of respectively unsaturated and saturated entries of $\boldsymbol{y}$. We note $\boldsymbol{A}^{\mathcal{I}}$ the matrix represents the row-submatrix of $A$ where only the rows indexed by $\mathcal{I}$ are selected.

Let $\mathcal{I}^{\boldsymbol{A}}_{\boldsymbol{x},\boldsymbol{u}} = \overline{\mathcal{I}_{\mathrm{sat}}}(\boldsymbol{Ax}) \cap \overline{\mathcal{I}_{\mathrm{sat}}}(\boldsymbol{Au})$ be the intersection of the unsaturated index sets of $\boldsymbol{Ax}$ and $\boldsymbol{Au}$. Using the law of total probability we can decompose the probability of Theorem 1 into the two target probabilities. By considering:

$$\mathcal{A} = \left\{ \exists \boldsymbol{x}, \boldsymbol{u} \in \mathcal{X}_R, \boldsymbol{x} \neq \boldsymbol{u} : \boldsymbol{A}^{\mathcal{I}^{\boldsymbol{A}}_{\boldsymbol{x},\boldsymbol{u}}} \boldsymbol{x} = \boldsymbol{A}^{\mathcal{I}^{\boldsymbol{A}}_{\boldsymbol{x},\boldsymbol{u}}} \boldsymbol{u} \right\}$$

$$\mathcal{B} = \left\{ \forall \boldsymbol{x}, \boldsymbol{u} \in \mathcal{X}_R, \boldsymbol{x} \neq \boldsymbol{u} : \left| \mathcal{I}^{\boldsymbol{A}}_{\boldsymbol{x},\boldsymbol{u}} \right| > (1-2\alpha)m \right\}$$

where $\mathcal{A}$ is the set of events where there exists two signals having the same common unsaturated measurements, $\mathcal{B}$ is the set of events where the number of common unsaturated measurements is greater than $(1-2\alpha)m$ for all couple of signals. And taking it into the law of total probability:

$$\begin{aligned}
\mathbb{P}(\mathcal{A}) &= \mathbb{P}(\mathcal{A} \cap \mathcal{B}) + \mathbb{P}(\mathcal{A} \cap \bar{\mathcal{B}}) \\
&= \mathbb{P}(\mathcal{A} \mid \mathcal{B}) \times \mathbb{P}(\mathcal{B}) + \mathbb{P}(\mathcal{A} \mid \bar{\mathcal{B}}) \times \mathbb{P}(\bar{\mathcal{B}}) \\
&\leq \mathbb{P}(\mathcal{A} \mid \mathcal{B}) + \mathbb{P}(\bar{\mathcal{B}})
\end{aligned} \tag{21}$$

as we have that $\mathbb{P}(\mathcal{B}) < 1$ and $\mathbb{P}(\mathcal{A} \mid \bar{\mathcal{B}}) < 1$.

$\mathbb{P}(\mathcal{A} \mid \mathcal{B})$ and $\mathbb{P}(\bar{\mathcal{B}})$ are respectively the probability that two signals have the same common unsaturated measurements given that the number of saturated measurements is low, and the probability there exist two distinct signals whose the number of common unsaturated measurements is low. Next, we upper-bound separately the two terms $\mathbb{P}(\mathcal{A} \mid \mathcal{B})$ and $\mathbb{P}(\bar{\mathcal{B}})$. We start with the first one:

$$\mathbb{P}(\mathcal{A} \mid \mathcal{B}) = \mathbb{P}\left( \left\{ \exists \boldsymbol{x}, \boldsymbol{u} \in \mathcal{X}_R, \boldsymbol{x} \neq \boldsymbol{u} : \boldsymbol{A}^{\mathcal{I}^{\boldsymbol{A}}_{\boldsymbol{x},\boldsymbol{u}}} \boldsymbol{x} = \boldsymbol{A}^{\mathcal{I}^{\boldsymbol{A}}_{\boldsymbol{x},\boldsymbol{u}}} \boldsymbol{u} \right\} \middle| \left\{ \forall \boldsymbol{x}, \boldsymbol{u} \in \mathcal{X}_R, \boldsymbol{x} \neq \boldsymbol{u} : \left| \mathcal{I}^{\boldsymbol{A}}_{\boldsymbol{x},\boldsymbol{u}} \right| > (1-2\alpha)m \right\} \right)$$

$$\leq \mathbb{P}\left( \exists\, \mathcal{I} \subset [\![0:m]\!], |\mathcal{I}| > (1-2\alpha)m, \ \exists \boldsymbol{x}, \boldsymbol{u} \in \mathcal{X}_R, \boldsymbol{x} \neq \boldsymbol{u} : \boldsymbol{A}^{\mathcal{I}} \boldsymbol{x} = \boldsymbol{A}^{\mathcal{I}} \boldsymbol{u} \right).$$

And because of $\mathcal{X}$ is a cone, we have (Robinson, 2010):

$$\dim(\mathcal{X}_R) \leq \dim(\mathcal{X} \cap \mathbb{S}) + 1 = \dim(S_{\mathcal{X}}) + 1 < k + 1$$

We thus have, as long as $(1-2\alpha)m > 2(k+1)$, and thanks to Theorem 3:

$$\mathbb{P}\left( \exists\, \mathcal{I} \subset [\![0:m]\!], |\mathcal{I}| > (1-2\alpha)m, \ \exists \boldsymbol{x}, \boldsymbol{u} \in \mathcal{X}_R, \boldsymbol{x} \neq \boldsymbol{u} : \boldsymbol{A}^{\mathcal{I}} \boldsymbol{x} = \boldsymbol{A}^{\mathcal{I}} \boldsymbol{u} \right) = 0 \tag{22}$$

And so the first right-hand side term of (21) is equals 0.

From $(1-2\alpha)m > 2(k+1)$ which imposes $\alpha < \frac{1}{2} - \frac{k+1}{m}$, we can fix $\alpha$ and therefore fix $R < \sqrt{m}\mu\left(\frac{1}{2} - \frac{k+1}{m}\right)$. We also note as $\alpha > 0$, we necessarily have $2(k+1) < m$.
We can now upper-bound the second term appearing in (21):

$$\mathbb{P}\left( \exists \boldsymbol{x}, \boldsymbol{u} \in \mathcal{X}_R, \boldsymbol{x} \neq \boldsymbol{u} : \left| \mathcal{I}^{\boldsymbol{A}}_{\boldsymbol{x},\boldsymbol{u}} \right| \leq (1-2\alpha)m \right).$$

By observing that for two set I and J of $\{1, \ldots, m\}$ with $|I \cap J| \leq (1-2\alpha)m$, this means that $\left| \bar{I} \cup \bar{J} \right| > 2\alpha m$, which implies that either $\left| \bar{I} \right| > \alpha m$ or $\left| \bar{J} \right| > \alpha m$ (since otherwise the union has size $< 2\alpha m$), therefore $|I| \leq (1-\alpha)m$ or $|J| \leq (1-\alpha)m$, we can deduce:

$$\begin{aligned}
&\left| \mathcal{I}^{\boldsymbol{A}}_{\boldsymbol{x},\boldsymbol{u}} \right| \leq (1-2\alpha)m \\
\Leftrightarrow\ &\left| \overline{\mathcal{I}_{\mathrm{sat}}}(\boldsymbol{Ax}) \cap \overline{\mathcal{I}_{\mathrm{sat}}}(\boldsymbol{Au}) \right| \leq (1-2\alpha)m \\
\Rightarrow\ &\begin{cases}
\left| \overline{\mathcal{I}_{\mathrm{sat}}}(\boldsymbol{Ax}) \right| \leq \left\lfloor \frac{m+(1-2\alpha)m}{2} \right\rfloor \leq (1-\alpha)m \\
\text{or} \\
\left| \overline{\mathcal{I}_{\mathrm{sat}}}(\boldsymbol{Au}) \right| \leq \left\lfloor \frac{m+(1-2\alpha)m}{2} \right\rfloor \leq (1-\alpha)m
\end{cases}
\end{aligned}$$

we have

$$\mathbb{P}\left(\exists \boldsymbol{x}, \boldsymbol{u} \in \mathcal{X}_R, \boldsymbol{x} \neq \boldsymbol{u} : |\mathcal{I}_{\boldsymbol{x},\boldsymbol{u}}^{\boldsymbol{A}}| \leq (1 - 2\alpha)m\right)$$
$$\leq \mathbb{P}\left(\left\{\exists \boldsymbol{x} \in \mathcal{X}_R : \left|\overline{\mathcal{I}_{\mathrm{sat}}}(\boldsymbol{A}\boldsymbol{x})\right| \leq (1 - \alpha)m\right\} \cup \left\{\exists \boldsymbol{u} \in \mathcal{X}_R : \left|\overline{\mathcal{I}_{\mathrm{sat}}}(\boldsymbol{A}\boldsymbol{u})\right| \leq (1 - \alpha)m\right\}\right)$$
$$\leq 2\mathbb{P}\left(\exists \boldsymbol{x} \in \mathcal{X}_R : \left|\overline{\mathcal{I}_{\mathrm{sat}}}(\boldsymbol{A}\boldsymbol{x})\right| \leq (1 - \alpha)m\right)$$

and if we remark that

$$\|\boldsymbol{A}\boldsymbol{x}\|_1 = \sum_{i=1}^m |\boldsymbol{a}_i^\top \boldsymbol{x}| \geq \sum_{i \in \mathcal{I}_{\mathrm{sat}}(\boldsymbol{A}\boldsymbol{x})} |\boldsymbol{a}_i^\top \boldsymbol{x}| \geq \sum_{i \in \mathcal{I}_{\mathrm{sat}}(\boldsymbol{A}\boldsymbol{x})} \mu$$

we obtain

$$\|\boldsymbol{A}\boldsymbol{x}\|_1 \geq |\mathcal{I}_{\mathrm{sat}}(\boldsymbol{A}\boldsymbol{x})|\,\mu.$$

Thanks to Corollary 2.1 we know that with probability at least $1 - 6e^{-Cm}$,

$$\|\boldsymbol{A}\boldsymbol{x}\|_1 \leq \sqrt{m}\|\boldsymbol{x}\|_2 \leq m\alpha\mu.$$

We deduce from these two last inequalities

$$|\mathcal{I}_{\mathrm{sat}}(\boldsymbol{A}\boldsymbol{x})|\,\mu \leq m\alpha\mu$$
$$m - \left|\overline{\mathcal{I}_{\mathrm{sat}}}(\boldsymbol{A}\boldsymbol{x})\right| \leq m\alpha$$
$$(1 - \alpha)m \leq \left|\overline{\mathcal{I}_{\mathrm{sat}}}(\boldsymbol{A}\boldsymbol{x})\right|.$$

And thus

$$\mathbb{P}\left(\exists \boldsymbol{x} \in \mathcal{X}_R : \left|\overline{\mathcal{I}_{\mathrm{sat}}}(\boldsymbol{A}\boldsymbol{x})\right| \leq (1 - \alpha)m\right) \leq 6e^{-Cm}$$

as long as

$$m \geq k \log\left(\frac{1}{\epsilon^*}\right).$$

Finally,

$$\mathbb{P}\left(\bar{\mathcal{B}}\right) \leq 12e^{-Cm}. \tag{23}$$

we conclude by adding the upper-bound obtain in (22) and (23):

$$\mathbb{P}\left(\exists \boldsymbol{x}, \boldsymbol{u} \in \mathcal{X}_R, \boldsymbol{x} \neq \boldsymbol{u} : \eta(\boldsymbol{A}\boldsymbol{x}) = \eta\left(\boldsymbol{A}\boldsymbol{u}\right)\right) \leq \mathbb{P}\left(\exists \boldsymbol{x}, \boldsymbol{u} \in \mathcal{X}_R, \boldsymbol{x} \neq \boldsymbol{u} : \boldsymbol{A}^{\mathcal{I}_{\boldsymbol{x},\boldsymbol{u}}^{\boldsymbol{A}}}\boldsymbol{x} = \boldsymbol{A}^{\mathcal{I}_{\boldsymbol{x},\boldsymbol{u}}^{\boldsymbol{A}}}\boldsymbol{u}\right)$$
$$\leq \mathbb{P}\left(\mathcal{A} \mid \mathcal{B}\right) + \mathbb{P}\left(\bar{\mathcal{B}}\right)$$
$$\leq 0 + 12e^{-Cm}.$$

Finally, (5) follows from a simple rescaling of the radius $R$ and the vectors $\boldsymbol{x}$ and $\boldsymbol{u}$ by a factor $\sqrt{m}$, and an opposite rescaling of the entries $A_{ij}$ of $\boldsymbol{A}$ by $1/\sqrt{m}$ so that $A_{ij} \sim \mathcal{N}(0, 1)$.

## B   Appendix: Hyperparameters, training details and images reconstruction

Table 5: Hyperparameters for different experiments.

| Section | $\lambda$ | $p_g$ | Architecture | Learning rate | Epochs | Batch size |
|---|---|---|---|---|---|---|
| Section 5.2.1 | 1 | $\mathcal{U}(0.1, 2)$ | Bias-free U-Net, 4 down + 4 up blocks | $5e^{-4}$ | 300 | 50 |
| Section 5.2.2 | 1 | $\mathcal{U}(0.5, 1.5)$ | MLP, input size 100, hidden size 100, depth 5 | $1e^{-4}$ | 300 | 100 |
| Section 5.3 | 0.1 | $\mathcal{U}(0.1, 2.0)$ | Bias-free U-Net, 5 down + 5 up blocks | $5e^{-4}$ | 100 | 35 |
| Section 5.4 | 0.1 | $\mathcal{U}(0.2, 1.5)$ | Bias-free U-Net, 4 down + 4 up blocks | $5e^{-5}$ | 400 | 12 |

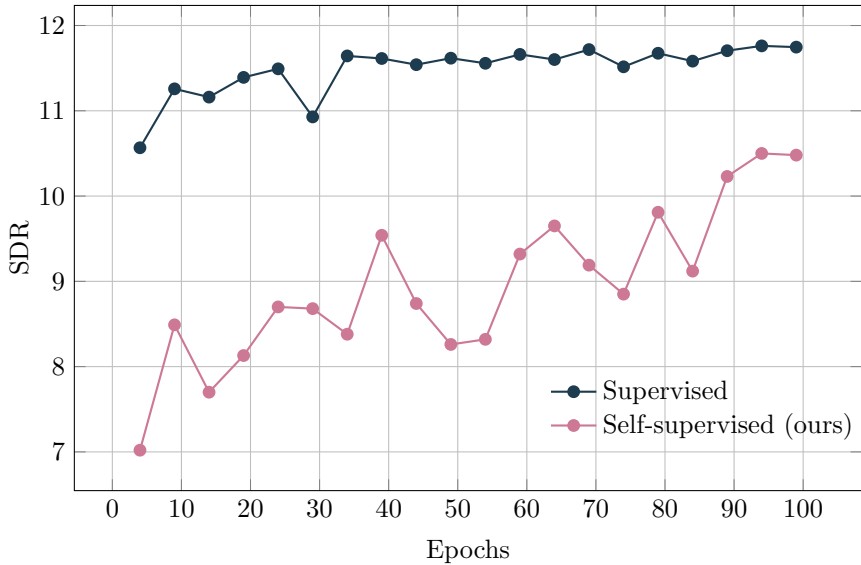

Figure 9: SDR performance on the evaluation dataset during training for supervised and self-supervised methods.

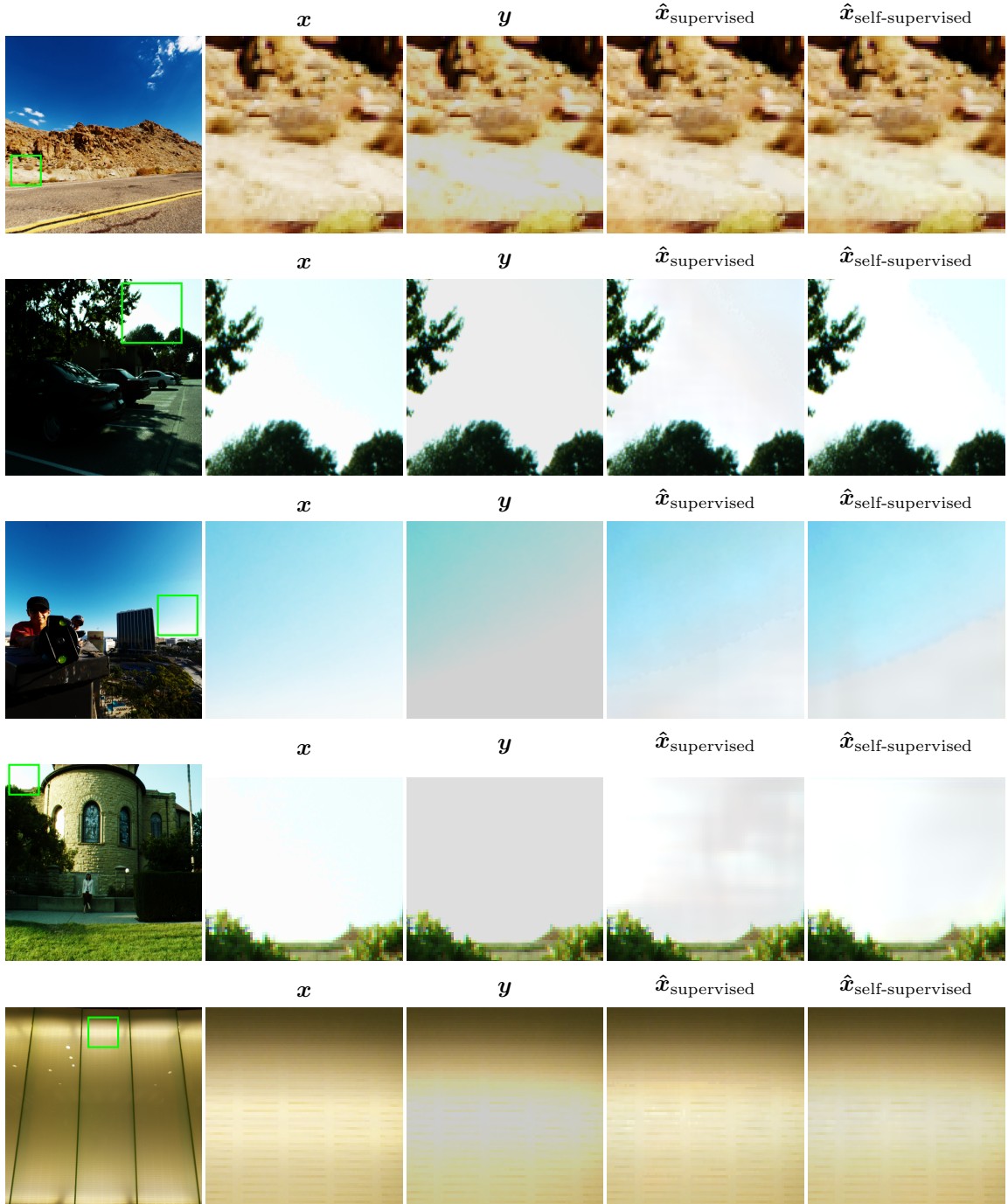

Figure 10: Image reconstruction on the test dataset. Artifacts can appear in saturation zone.

