# OpenReview forum: "Learning to reconstruct from saturated data: audio declipping and high-dynamic range imaging"
_TMLR — Accepted by TMLR_

### Review · Reviewer_SbLd · 2025-09-19

**Summary Of Contributions:**

This work considers the nonlinear inverse problem with saturated measurements, under the assumption that the underlying signal set is conical. This manifests as a special instance of equivariant learning. Using the same network architecture, the authors demonstrate that the proposed self-supervised method is reasonably well performing compared to supervised training, with the expected additional computational burden.

**Audience:**

Yes

**Audience Explanation:**

* The authors consider an interesting nonlinear problem which is not entirely covered by existing theory.
* Theorem 1 is interesting as it considers the interaction between random feature and saturation. It provides a vanishing bound on the probability that two data will result in the same measurement, based on the box-counting/intrinsic dimension of the underlying signal set.

**Claims And Evidence:**

Yes

**Claims Explanation:**

* The experimental section is sufficiently ablated and clearly demonstrates the effect of each of the authors' modifications. The imaging section also compares to a powerful pretrained method (DPIR), and shows that the different degradation operator needs to be separately addressed.

**Requested Changes:**

Strengthening points

* Minor formatting/clarity issues that can be fixed easily. I list some easily fixable ones in the following section.
	* Example 1: maybe state that no additional priors are required on $\mathcal{X}$.
	* Prop 1 is strangely placed and may be bit confusing by itself. Maybe combine with Remark 1?
* Prop 3 can be simplified even more. Inscribing a sphere of radius $\mu$ into the hypercube, the Euclidean projection of the hypercube onto the sphere is bijective. Then you can use that the cone is uniquely defined by its intersection with the sphere of radius $\mu$ (and therefore by its intersection with the hypercube).
* "proposed self-supervised with bias": I could not find the definition of this, but I assume it is the UNet but without the modification in 4.2.2?
* Are there assumptions on $m,n$ in Theorem 1? It would be useful to have this in the statement.
* Section 4.1 could do with some relations to variational regularization, in this case the regularization corresponding to the scale-invariance prior is given by the equivariance loss (11).
* Fig 3 needs more detail in the caption, state that removing the bias makes the network naturally 1-homogeneous (... logic here ... ) and improves the reconstruction.
* How does $C$ in Theorem 1 depend on $R$? This would be nice to quantify the effect of clipping.

**Typos/small changes**
- (p.1) "mispecified" -> "misspecified", "the inverse operator" -> "an inverse operator"
- (p.2) could you clarify how self supervised learning mitigates distribution shift?
- (p.3) reference issue in (Moliner et al., 2023), it starts as a new sentence. Also it may be good to add some examples of HDR image artifacts, like bars in low brightness gradient areas, or overblown whites
- (p.3) "for the case **of** Gaussian measurements and **using** an $\ell_1$ minimization approach"
- (p.4) when defining model identification: perhaps "for a forward operator $\eta$ and measurement sets $\mathcal{Y}$, does a unique signal set $\mathcal{X}$ exist satisfying some mild priors and $\mathcal{Y} = \eta(\mathcal{X})$", or something along those lines. Also should it be $\mathcal{Y}$ instead of $\{\mathbf{y}_i\}$?
- (p.4) Reference issue (Chen et al.). "group of transformations $G$ for which the ..." -> "...transformations $G$ under which the ..."
- (p.6) "as it is common signal recovery analyses" -> "as commonly employed in signal recovery analyses" . Also random Gaussian matrix is not a rotation, maybe approximate rotation.
- (p.7) "here $\mathcal{Z}$ is a cone" -> "where ..."
- (p.13) "mintutes"
- (p.15) "for $\varphi$ and $\psi$ two generic functions" -> "for two closed proper convex functions $\varphi$ and $\psi$"
- Equations are sometimes referred to as Equation (xx) or sometimes just (xx). Please unify

---

> ### Author Response · Authors · 2025-10-30
> **Response to the reviews part 1.**
>
> Thank you for your feedback. All the changes in the updated manuscript are colored in blue (earlier modifications being kept in red). Below is a response to each of your points.
>
> >Prop 1 is strangely placed and may be bit confusing by itself. Maybe combine with Remark 1?
>
> Thank you for your comment. We have decided to replace the proposition with a sentence introducing the following paragraph which explains why we need more hypotheses about X.
>
> > Prop 3 can be simplified even more. Inscribing a sphere of radius $\mu$ into the hypercube, the Euclidean projection of the hypercube onto the sphere is bijective. Then you can use that the cone is uniquely defined by its intersection with the sphere of radius $\mu$ (and therefore by its intersection with the hypercube).
>
> Thank you for this elegant demonstration. Nevertheless, we believe that a bijection only applies to the projection of the frontier of the hypercube onto a sphere. To take full advantage of the solution proposed by this reviewer, we would first have to prove that the key hypercube part lies indeed on its boundary, which would not shorten the proof. As we believe our proof is perhaps more constructive, we prefer to keep it as it is.
>
> > "proposed self-supervised with bias": I could not find the definition of this, but I assume it is the UNet but without the modification in 4.2.2?
>
> Yes, that is right. We have therefore added a sentence to the paragraph describing the results to mention the proposed method with bias, which refers to section 4.2.2. We hope this clarifies the situation.
>
> > Are there assumptions on $m,n$  in Theorem 1? It would be useful to have this in the statement.
>
> Following a comment from reviewer 9rV1, we have reworded the theorem and taken the opportunity to add “for m, n>0.” Note we can find next conditions on $m$ relative to $k$:
> $$
> \textstyle m \geq k\log(1/\epsilon^*)\ \text{and}\ R< \mu(\frac{1}{2} - \frac{k+1}{m}),
> $$
> which implies $R$ does not exist if $k=0$ and $m\leq 2$.
>
>
> > Section 4.1 could do with some relations to variational regularization, in this case the regularization corresponding to the scale-invariance prior is given by the equivariance loss (11).
>
> Yes, you are right, it is related, but since the constraint is on $f$ and not on $x$, there is a certain difference with variational regularization. We are therefore concerned that linking the two methods may add confusion to the reader.
>
> > Fig 3 needs more detail in the caption, state that removing the bias makes the network naturally 1-homogeneous (... logic here ... ) and improves the reconstruction.
>
> We added the following sentence: "We thus observe that removing the bias makes the network naturally homogeneous and improves the reconstruction." in the caption.
>
> > How does $C$ in Theorem 1 depend on $R$? This would be nice to quantify the effect of clipping.
>
> The constant $C$ is an absolute constant and, therefore, does not depend on $R$. It comes from the theorem 2 (Puy et al. 2017, Theorem II.2).
>
> > (p.2) could you clarify how self supervised learning mitigates distribution shift?
>
> In real-life applications where ground truth are unavailable, there are no predefined training or testing datasets—only the raw signals that need reconstruction. In such cases, the model can be trained directly on the measurement dataset itself.
>
> > Also it may be good to add some examples of HDR image artifacts, like bars in low brightness gradient areas, or overblown whites
>
> We added in appendix five new examples. In some of them, we can see artifacts due to saturation.
>
>
> > (p.4) when defining model identification: perhaps "for a forward operator $\eta$ and measurement sets $Y$, does a unique signal set $X$ exist satisfying some mild priors and $Y=\eta(X)$", or something along those lines. Also should it be $Y$ instead of $y_i$?
>
> Thank you for this comment, which we have taken into account by modifying the article. The new sentence become: _``for a forward operator $\eta$ and measurement sets $Y$, does a unique signal set $X$ exist satisfying some mild priors and $Y=\eta(X)$? In other words, we want to know if it is possible to find the support of the signal distribution $X$ from measurement data $\left\{ y_i \right\}^N_{i=1}$''_
>
>
> >Equations are sometimes referred to as Equation (xx) or sometimes just (xx). Please unify
>
> Thank you, we have now uniformized our notations.

---

> ### Author Response · Authors · 2025-10-30
> **Response to the reviews part 2.**
>
> We have addressed all of the following points and thank you for bringing them to our attention:
> - (p.1) "mispecified" -> "misspecified", "the inverse operator" -> "an inverse operator"
> - (p.3) reference issue in (Moliner et al., 2023), it starts as a new sentence.
> - (p.3) "for the case **of** Gaussian measurements and **using** an $l_1$ minimization approach"
> - (p.4) Reference issue (Chen et al.).
> - (p.4) "group of transformations $G$ for which the ..." -> "...transformations $G$ under which the ..."
> - (p.6) "as it is common signal recovery analyses" -> "as commonly employed in signal recovery analyses" . Also random Gaussian matrix is not a rotation, maybe approximate rotation.
> - (p.7) "here $Z$ is a cone" -> "where ..."
> - (p.13) "mintutes"
> - (p.15) "for $\varphi$ and $\psi$ two generic functions" -> "for two closed proper convex functions $\varphi$ and $\psi$"
> - Example 1: maybe state that no additional priors are required on X

---

### Review · Reviewer_UfYi · 2025-10-01

**Summary Of Contributions:**

This paper presents a self-supervised framework for reconstructing signals from saturated measurements, addressing nonlinear inverse problems in audio declipping and high dynamic range (HDR) imaging. The authors identify and formalize amplitude invariance as a key property enabling self-supervised learning in such settings. The paper makes the following contributions:

Theoretical Foundations: The authors provide sufficient conditions under which a signal model can be identified and signals can be recovered solely from saturated observations. These conditions rely on the assumption that the signal set is scale-invariant (i.e., conic) and has low box-counting dimension.

Self-supervised Loss Design: A novel self-supervised loss is proposed, combining:

A measurement consistency loss tailored to handle non-zero gradients even in the presence of clipping;

An equivariance loss that enforces consistency under amplitude scaling, reflecting the scale invariance of the data.

Experiments on Audio and Images: The method is validated on both synthetic and real-world data:

In audio declipping, the self-supervised method outperforms traditional methods and approaches the performance of fully supervised models.

In HDR imaging, it performs comparably to supervised baselines, despite not requiring ground-truth during training.

Overall, the work significantly expands the scope of self-supervised learning in inverse problems beyond linear regimes by handling nonlinear saturations, a common real-world issue.

**Audience:**

Yes

**Audience Explanation:**

The paper addresses a timely and relevant problem in machine learning, and its findings would be of interest to researchers and practitioners in the field. The proposed approach and results contribute to ongoing discussions in the community, making the paper suitable for TMLR’s readership.

**Broader Impact Concerns:**

No significant ethical risks were identified in this work.

**Claims And Evidence:**

Yes

**Claims Explanation:**

The submission provides sufficient experimental results and comparisons with relevant baselines to support its main claims. The methodology is described clearly, and the evidence is presented in a structured and convincing manner.

**Requested Changes:**

The author has already responded well to my previous questions, and I have no further questions.

---

> ### Author Response · Authors · 2025-10-30
> **Response to the reviews.**
>
> We thank you once again for your review and the previous comments you have given us.

---

### Review · Reviewer_9rV1 · 2025-10-25

**Summary Of Contributions:**

This review is to reconsider the submission of the same paper as previous submitted to TMLR, with reference link [here](https://openreview.net/forum?id=gwDNM3b353).

Looking at the current version, there is literally no change from the previously rejected version. In particular, the main issue:
```
(1) The main theoretical result presented in the manuscript is trivial after authors reformulated proposition 3. Theorem 1 is also within expectation, because when the variance of the random sampling/compression matrix A is small (i.e. when m>>1), the approximate isometry of Gaussian matrix has classic Johnson-Lindenstrauss lemma that serves as the prototype to the presented result, regardless of the inclusion of the thresholding operator.

(2) As another reviewer pointed out, there is no essential connection between the presented theorem and the practical reconstruction algorithm (the application of the invariance penalty).

The measure consistency loss does not make much sense. It reads that the thresholded reconstructed signal should align with the observation. Then there are infinite many solutions to this problem.

(3) Authors still fail to formulate the problem they try to solve as a clear optimisation problem.
```

None of these points is addressed. As there is no change to the previous version, the conclusion should be the same.

**Audience:**

Yes

**Audience Explanation:**

Topic is relevant

**Claims And Evidence:**

No

**Claims Explanation:**

See summary section.

**Requested Changes:**

N/A

---

> ### Author Response · Authors · 2025-10-30
> **response to the reviews part1.**
>
> Thank you for your feedback. All the changes in the updated manuscript are colored in blue (earlier modifications being kept in red). Below is a response to each of your points.
>
> > Theorem 1 is within expectation, because when the variance of the random sampling/compression matrix A is small (i.e. when m>>1), the approximate isometry of Gaussian matrix has classic Johnson-Lindenstrauss lemma that serves as the prototype to the presented result, regardless of the inclusion of the thresholding operator.
>
> We thank this reviewer for pointing out again this observation, which we had not properly understood. Indeed, observing that the variance of $A$ decreases as $1/m$ can lead to disregarding the impact of saturation $\eta$ for large values of $m$. However, the radius $R$ of the restricted signal set $\mathcal X_R$  (bounding the maximum norm of the signals) increases at the same time with $\sqrt m$; the interaction between the variance of the matrix and the radius of the signal set was therefore somewhat difficult to disentangle. Using a simple recalibration argument applied to $A$, $x$, $u$ and $R$, we therefore reformulated Theorem 1 to consider a matrix $A$ with unit variance and an almost constant radius $R$. We believe that this simplifies the reading of the theorem statement; in particular, it shows that **the fraction of saturated measurements in $y$ is constant as a function of $m$**. Thus, it is not possible to remove the impact of the non-linearity for large values of $m$ when proving the theorem. In the spirit of a certain literature dedicated to one bit and quantized compressive sensing, tools other than the classical Johnson-Lindenstrauss lemma (or the restricted isometry property) must be used to understand the possible injectivity of $\eta \circ A$ on $\mathcal X_R$, which is the objective of this theorem. We hope that these modifications, based on this reviewer's comment, will make the statement of the theorem and its implications clearer.
>
> > The main theoretical result presented in the manuscript is trivial after authors reformulated proposition 3.
>
> We are still emphasizing that proposition 3 is not our main result—that distinction belongs to Theorem 1, which justifies our choice of terminology. Proposition 3 demonstrates that scale invariance is essential for learning from clipped data, a finding we confirm experimentally on both audio and image signals. We consider it both relevant and necessary to state explicitly, despite its straightforward proof.
>
> > As another reviewer pointed out, there is no essential connection between the presented theorem and the practical reconstruction algorithm (the application of the invariance penalty).
> > The measure consistency loss does not make much sense. It reads that the thresholded reconstructed signal should align with the observation. Then there are infinite many solutions to this problem.
>
> We agree that the measurement-consistency (MC) loss alone does not make sense, just as the equivariant-imaging (EI) loss alone would not make sense either (since it would not lead to a reconstruction consistent with the measurements, i.e., $h(f(y))\approx y$). The idea is to combine both losses to obtain reconstructions that are i) coherent with the measurements and that ii) belong to a conic (amplitude invariant) signal set. Indeed, the theoretical part guarantees that there is a unique signal set given the measurement set, under the assumption of invariance to amplitude. Therefore, the reconstructions should belong to this set, since we are exploiting the same assumption to train the neural network.
> To illustrate this idea, we had added in Table 1 (included here below as well), the performance for $\mathcal{L}_{\textrm{MC}}$ alone to compare with the proposed method. Using both losses performs significantly better than using the MC loss alone, reaching a performance that is close to the supervised loss.
> | Methods                                                                          |  SDR $\uparrow$  |
> | :------------------------------------------------------------------------------- | :--------------: |
> | Identity                                                                         | $4.84 \pm 2.12$  |
> | Social Sparsity                                                                  | $9.92 \pm 4.46$  |
> | Supervised                                                                       | $11.69 \pm 2.25$ |
> | Proposed self-supervised $\mathcal{L}_{\textrm{MC}} + \mathcal{L}_{\textrm{EI}}$ | $10.48 \pm 2.20$ |
> | Self-supervised with $\mathcal{L}_{\textrm{MC}}$ alone                              | $4.84 \pm 2.12$  |
> We did not include result for the loss $\mathcal{L}_{\textrm{EI}}$ alone because non-measurement consistent results would be irrelevant.

---

> ### Author Response · Authors · 2025-10-30
> **response to the reviews part 2.**
>
> > Authors still fail to formulate the problem they try to solve as a clear optimisation problem.
>
> We assume here that the reviewer would like us to follow standard practices in the literature of variational optimization for inverse problems, where the focus is generally on recovering *a single signal* from multiple observations by formulating an optimization problem whose result provides an estimate of the signal. However, to the best of our knowledge such a framework is not applicable to the problem of *identifying a whole set of signals* (a model identification question) when only their (saturated) observations are available. We would like to point out that other works in the field of self-supervised learning also do not write the model identification problem as an optimization one, see for example
> - Julián Tachella and Laurent Jacques. Learning to reconstruct signals from binary measurements. In TMLR (featured certification), 2024.
>
> More generally, other fundamental questions in the context of supervised learning, such as the separability of signal classes after linear dimensionality reduction, are also not formulated as an optimization problem, see for example
> - A. Bandeira, D. Mixon, B. Recht. Compressive classification and the rare eclipse problem. In Compressed Sensing and its Applications: Second International MATHEON Conference 2015 (pp. 197-220). Cham: Springer International Publishing.

---

### Decision · Action_Editor_n86w · 2026-01-04

**Recommendation:** Accept with minor revision

**Additional Comments:**

The authors should either present the results in Sections 3 and 4/5 as independent, or make a stronger connection between the algorithmic choices in Section 4 and the theoretical development in Section 3. They are not linked, except that they both address the clipping problem from different viewpoints.

**Audience:**

Yes

**Audience Explanation:**

The paper presents a self-supervised solution for the inverse problem of reconstructing a clipped sequence. The idea that this problem is amenable to self-supervised learning is novel and may prompt further investigation into self-supervised learning algorithms.

**Claims And Evidence:**

Yes

**Claims Explanation:**

The paper contains two parts: one that shows how many samples are required to reconstruct a clipped image or audio signal given the dynamic range. The second part proposed a self-supervised reconstruction task that is competitive with a supervised approach.

I prefer papers with a single message, and the theoretical results in Section 3 are not used in the algorithm proposed in Section 4. I do not see a connection between Theorem 1 and the losses in equations 7 and 11.

The paper presents a self-supervised method that solves the problem and performs comparably to a supervised learning algorithm, which is noteworthy. However, it will be good to add in which cases this might be a relevant problem, as I can imagine that if the problem is appropriate, because a low-quality device is being used to measure a signal that might be saturated, why can we not use the high-quality device to train the supervised solution in the first place?

---

> ### Author Response · Authors · 2026-01-29
> **Response and Paper Updates**
>
> Dear Area Chair,
>
> Thank you for your comments and recommendations on our work.
>
> We have updated the manuscript according to your recommendations. Specifically:
> - We have added a sentence to the 'Related Work' Section to clarify why using a camera with a larger dynamic range to obtain ground-truth references for supervised learning is often infeasible.  We illustrate this with an example of ultra-high speed imaging of combustion phenomena, where obtaining ground-truth references is impossible.
> - We have added a sentence in Section 3.1 clarifying the connection between the theoretical contributions in Section 3 and the practical self-supervised loss in Section 4. The proposed theoretical framework allows us to determine the feasibility of learning to reconstruct from saturated data alone, which requires having enough observations for both model identification (i.e., learning the signal distribution) and signal recovery (i.e., reconstructing a signal given knowledge of the distribution).